# Modular actin nano-architecture enables podosome protrusion and mechanosensing

Koen van den Dries[1], Leila Nahidiazar[2,3,9], Johan A. Slotman[4,9], Marjolein B.M. Meddens[5], Elvis Pandzic[6], Ben Joosten[1], Marleen Ansems[7], Joost Schouwstra[1], Anke Meijer [1], Raymond Steen [1], Mietske Wijers[1], Jack Fransen[1], Adriaan B. Houtsmuller[4], Paul W. Wiseman [8], Kees Jalink[2,3] & Alessandra Cambi [1]*

Basement membrane transmigration during embryonal development, tissue homeostasis and tumor invasion relies on invadosomes, a collective term for invadopodia and podosomes. An adequate structural framework for this process is still missing. Here, we reveal the modular actin nano-architecture that enables podosome protrusion and mechanosensing. The podosome protrusive core contains a central branched actin module encased by a linear actin module, each harboring specific actin interactors and actin isoforms. From the core, two actin modules radiate: ventral filaments bound by vinculin and connected to the plasma membrane and dorsal interpodosomal filaments crosslinked by myosin IIA. On stiff substrates, the actin modules mediate long-range substrate exploration, associated with degradative behavior. On compliant substrates, the vinculin-bound ventral actin filaments shorten, resulting in short-range connectivity and a focally protrusive, non-degradative state. Our findings redefine podosome nanoscale architecture and reveal a paradigm for how actin modularity drives invadosome mechanosensing in cells that breach tissue boundaries.

[1] Department of Cell Biology, Radboud Institute for Molecular Life Sciences, Radboud University Medical Center, Nijmegen, Netherlands. [2] Division of Cell Biology, The Netherlands Cancer Institute, Amsterdam, Netherlands. [3] van Leeuwenhoek Centre of Advanced Microscopy, Amsterdam, Netherlands. [4] Department of Pathology, Optical imaging center Erasmus MC, Rotterdam, Netherlands. [5] Department of Physics and Astronomy and Department of Pathology, University of New Mexico, Albuquerque, NM 87131, USA. [6] Biomedical Imaging Facility, Mark Wainwright Analytical Centre, University of New South Wales, Sydney, NSW 2052, Australia. [7] Radiotherapy & OncoImmunology Laboratory, Department of Radiation Oncology, Radboud University Medical Center, Nijmegen, Netherlands. [8] Departments of Physics and Chemistry, McGill University Otto Maass (OM), Chemistry Building, 801 Sherbrooke Street West, Montreal, QC H3A 0B8, Canada. [9] These authors contributed equally: Leila Nahidiazar, Johan A. Slotman. *email: Alessandra. Cambi@radboudumc.nl

Cell–cell and cell–matrix interactions are controlled by actin-based machineries, such as adherens junctions, focal adhesions, and invadosomes[1–3]. Recent insights into the nanoscale architecture of adherens junctions[4] and focal adhesions[5] have significantly furthered our mechanistic understanding of cell–cell interactions in organ epithelia and of cell–matrix interactions in cells that crawl through interstitial tissue, respectively. Much less defined, however, are the mechanisms that regulate the cytoskeletal organization in cells that carry out basement membrane transmigration or bone remodeling[6,7], which relies on the focal degradation and protrusion by invadosomes, a collective term for invadopodia and podosomes[3].

Invadosome-mediated basement membrane transmigration is a key process during development and tissue homeostasis. During *Caenorhabditis elegans* embryonic development, an anchor cell deploys invadopodia to breach the basement membrane separating the uterine and vulval epithelium[8]. To control tissue homeostasis, megakaryocytes use podosomes for shedding platelets into the bloodstream[9], endothelial cells for initiating new vessel sprouts[10] and leukocytes for leaving or entering blood vessels[11] and facilitating antigen capture[12]. Furthermore, podosome-mediated bone remodeling by osteoclasts is essential for proper bone homeostasis[13,14]. Finally, during tumorigenesis, cancer cells assemble invadopodia to initiate cell invasion, one of the first steps towards cancer metastasis[15]. Unravelling the basic mechanisms that control invadosome-mediated protrusion and environment probing enhances our understanding of these invasive processes.

Podosomes are characterized by a protrusive actin-rich core (500–700 nm) which is surrounded by an adhesive ring (200–300 nm) enriched for adaptor proteins, such as vinculin and talin[16]. Neighboring podosomes are interconnected by a network of bundled actin filaments that radiate from the podosome core and facilitate a mesoscale (1.5–10 μm) connectivity[17–19]. While individual podosomes are thought to function as micron-sized protrusive machineries[20–22], their mesoscale connectivity facilitates long-range basement membrane exploration for protrusion-permissive spots[18,23]. An adequate structural framework, however, that explains podosome protrusion and mechanosensing is still lacking. Also, how podosome mechanosensing relates to podosome mesoscale connectivity and degradative capacity remains elusive.

Using super-resolution–microscopy in both fixed and living primary human dendritic cells (DCs), we here reveal a modular actin nano-architecture that explains podosome protrusion and mechanosensing. We find that the podosome core consists of a two-module actin assembly with a central protrusion module (cPM) of branched actin filaments encased by linear actin filaments forming a peripheral protrusion module (pPM). We also show that the interpodosomal actin filaments that radiate from the core comprise a ventral module, bound by the cytoskeletal adapter protein vinculin, and a dorsal module, crosslinked by myosin IIA. Super-resolution microscopy and spatiotemporal image correlation spectroscopy on substrates with different stiffness revealed that on stiff substrates, podosomes mediate long-range substrate exploration, and a degradative behavior while on soft substrates, the ventral actin filaments become less prominent, resulting in short-range connectivity and an associated focally protrusive, non-degradative state. Our findings redefine the podosome nanoscale architecture and show how actin modularity enables invadosome mechanosensing in cells that breach tissue boundaries.

## Results

### Actin-binding proteins localize to distinct core submodules.
Actin-binding proteins such as WASP, arp2/3, cortactin, and α-actinin locate to podosomes cores in macrophages and rat smooth muscle cells[24–26]. While WASP, arp2/3, and cortactin primarily associate with branched actin[27,28], α-actinin primarily associates with linear actin filaments[29,30]. We therefore hypothesized that these actin-binding proteins may localize to different, spatially separated, regions within the podosome core. To investigate this, we examined and quantified the localization of these proteins with respect to actin.

We first examined the localization of WASP and arp3 by conventional fluorescence microscopy and observed that, also in DCs, these proteins localize to the podosome core (Fig. 1a, b). Interestingly, radial fluorescence profile analysis of hundreds of individual podosomes (Supplementary Fig. 1) revealed that the fluorescence signal from these proteins is confined to an area that is significantly smaller than the actin fluorescence area (Fig. 1a, b). Calculating the full width at half maximum (FWHM) of the intensity profiles indicated that the area to which the branched actin-binding proteins localize is approximately half the size of the total actin area, i.e. 0.38 ± 0.09 μm for WASP and 0.75 ± 0.28 μm for actin (Fig. 1a) and 0.40 ± 0.15 μm for arp3 and 0.69 ± 0.17 μm for actin (Fig. 1b). The branched actin-binding proteins thus appear to only occupy the most central part of the podosome core, a region we here term the cPM.

Next, we examined the localization of α-actinin by conventional fluorescence microscopy. Again, we observed a clear co-localization of α-actinin with the podosome core, but radial fluorescence profile analysis this time revealed that α-actinin localizes to a well-defined region at the core periphery (Fig. 1c). To study the localization of α-actinin in greater detail, we performed 3D-structured illumination super-resolution microscopy (3D-SIM) and confirmed our initial observation that α-actinin predominantly localizes to the core periphery (Fig. 1d, f). More importantly, 3D-SIM analysis also revealed that α-actinin localizes to a dome-shaped region at the core, a region we here term the pPM (Fig. 1d, e). Quantification of the α-actinin fluorescence profiles obtained with 3D-SIM indicated that the thickness of the pPM is 0.40 ± 0.10 μm (as measured by the FWHM, Fig. 1g) and its diameter 0.77 ± 0.25 μm (Fig. 1h), the latter being similar to the actin FWHM reported above (~0.75 μm, Fig. 1a, b). Interestingly, at the ventral part of podosomes, α-actinin partially colocalizes with vinculin (Fig. 1d, e), indicating that the pPM is closely associated with the integrins.

To confirm the differential localization of the actin-binding proteins in living cells, we co-transfected DCs with cortactin-BFP, vinculin-GFP, α-actinin-tagRFP, and Lifeact-iRFP and performed four color live-cell imaging by conventional microscopy (Supplementary Fig. 2 and Supplementary Movie 1). Also in living cells, two distinct protrusion modules could be discerned, with a cPM enriched for cortactin and a pPM enriched for α-actinin, fully supporting our observations in fixed cells.

**β-actin and γ-actin differentially localize to cPM and pPM.** In non-muscle cells, branched filaments mostly consist of β-actin, while linear filaments mostly consist of γ-actin[31,32]. We therefore investigated the localization of β and γ-actin in podosomes by Airyscan super-resolution microscopy. Interestingly, we observed a preferential localization of β-actin to the cPM and of γ-actin to the pPM (Fig. 2a, b). To note, the network of actin filaments in between podosomes primarily consist of γ-actin (Fig. 2a). 3D analysis revealed that γ-actin surrounds β-actin in the podosome core (Fig. 2a, b). Quantification of γ-actin fluorescence profile indicated a pPM thickness of 0.48 ± 0.16 μm and a diameter of 0.75 ± 0.18 μm, which corresponds very well with the values for α-actinin, indicating that both occupy the pPM (Fig. 2c, d). Application of 3D stochastic optical reconstruction microscopy

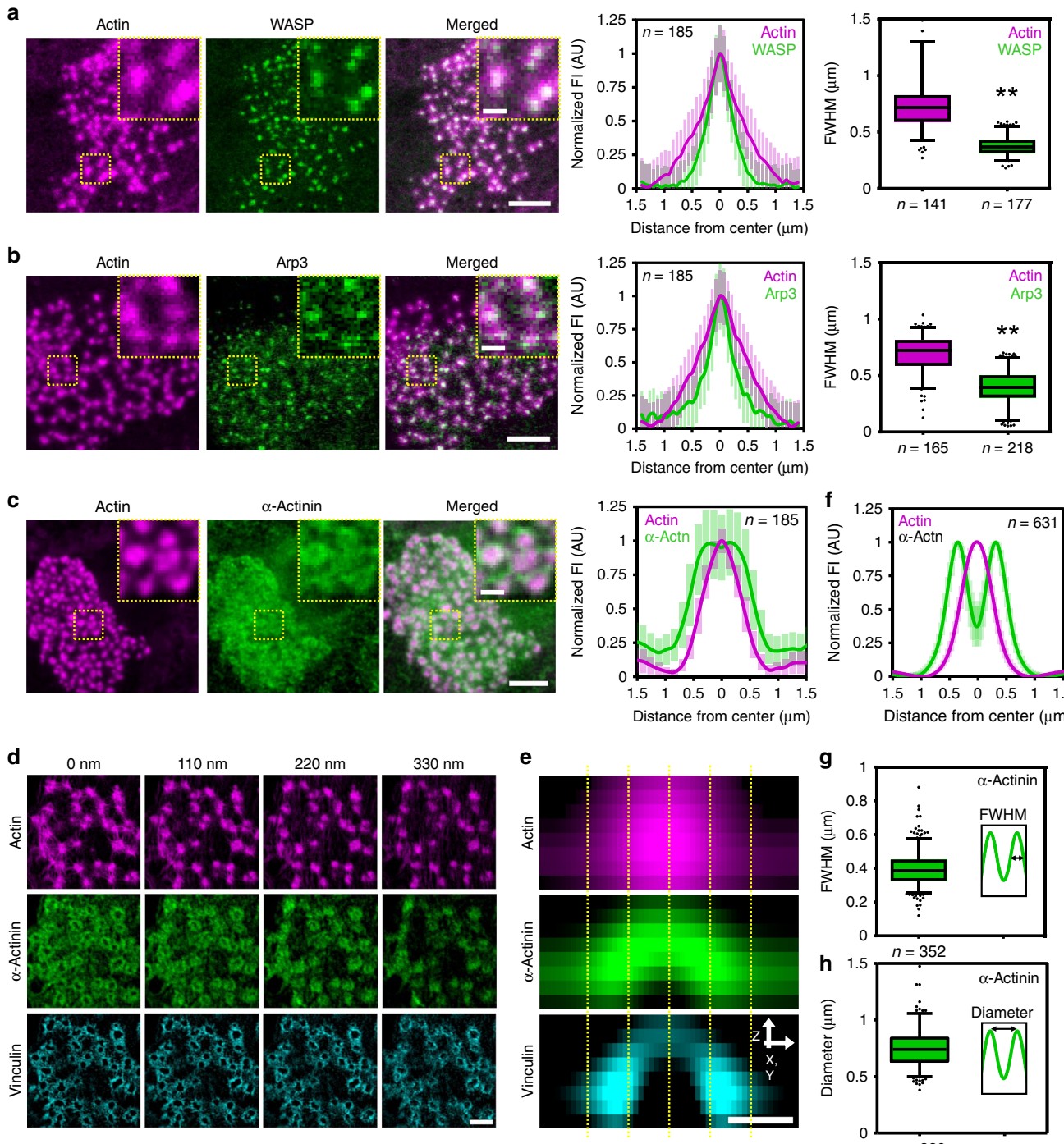

**Fig. 1** Actin-binding proteins in protrusive core differentially localize to podosome submodules. **a** Confocal images of a DC transfected with WASP-GFP (green) and stained for actin (magenta). The insets depict a few individual podosomes. The left graph shows the average ± s.d. radial fluorescent intensity profile of actin and WASP (n = 185 podosomes). The right graph depicts the FWHM of the fluorescent profile of actin (n = 141 podosomes) and WASP (n = 177 podosomes) pooled from three independent experiments. Statistical analysis was performed with an unpaired two-tailed Student's t-test. **P < 0.01. **b** Confocal images of a DC transfected with Arp3-GFP and stained for phalloidin to visualize actin (magenta). The insets depict a few individual podosomes. The left graph shows the average ± s.d. radial fluorescent intensity profile of actin and Arp3 (n = 185 podosomes). The right graph depicts the FWHM of the fluorescent profile of actin (n = 165 podosomes) and Arp3 (n = 218 podosomes) pooled from three independent experiments. Statistical analysis was performed with an unpaired two-tailed Student's t-test. **P < 0.01. **c** Confocal images of a DC stained for α-actinin (green) and actin (magenta). The insets depict a few individual podosomes. The graph shows the average ± s.d. radial fluorescent intensity profile of α-actinin and actin (n = 185 podosomes pooled from two independent experiments). **d** 3D-SIM images of a DC transfected with α-actinin-HA and stained for HA (green), actin (magenta), and vinculin (cyan). **e** Average radial orthogonal view of actin, α-actinin, and vinculin (n = 180 podosomes). **f** Average ± s.e.m. radial fluorescent intensity profile of actin and α-actinin obtained from the SIM images (at z:110 nm) (n = 631 podosomes pooled from three independent experiments). **g** Quantification of the FWHM of the α-actinin fluorescent profile (n = 352 podosomes pooled from three independent experiments). **h** Quantification of the α-actinin ring diameter (n = 280 podosomes pooled from three independent experiments). Scale bars: **a–c** = 5 μm, **d** = 1 μm, **e** = 0.5 μm; insets: **a–c** 1 μm. FI fluorescent intensity, AU arbitrary units. Box plots indicate median (middle line), 25th, 75th percentile (box) and 5th and 95th percentile (whiskers) as well as outliers (single points). Source data are provided as a Source Data file

(3D-STORM) gave similar results, indicating that our observations were not influenced by the resolution and deconvolution algorithm of the Airyscan approach (Fig. 2e, f).

To investigate whether the differential organization of the actin isoforms is a common feature of DC podosomes, we labeled murine bone marrow-derived DCs (BMDCs) for β and γ-actin (Supplementary Fig. 3). Also here, we found a cPM enriched for β-actin and a pPM enriched for γ-actin, demonstrating that the differential distribution of the actin isoforms is a common and conserved feature.

Altogether, these results demonstrate that within the 700 nm large podosome core, two distinct actin modules exist (Fig. 2g): a branched β-actin-rich central module (the cPM), where also WASP, cortactin, and Arp2/3 are found, and a linear γ-actin-rich peripheral module (the pPM), which completely encases the cPM and is crosslinked by α-actinin and partially bound by vinculin.

**Myosin IIA crosslinks dorsal interpodosomal actin filaments.** Myosin IIA is known to be associated with interpodosomal filaments and we and others demonstrated its role in regulating podosome dynamics and dissolution[18,22,33,34]. We showed previously that blocking myosin IIA activity with blebbistatin arrested podosome pushing behavior and mesoscale coordination, but the organization of the mechanosensitive proteins zyxin and vinculin remained unaltered[17]. This suggested that myosin IIA activity and mechanosensation were uncoupled at podosomes, and we therefore now sought to investigate whether myosin IIA and vinculin could occupy distinct filaments by performing 3D Airyscan imaging on DCs labelled for actin, vinculin, and myosin IIA (Fig. 3a, Supplementary Fig. 4). Visual inspection showed two striking differences in the localization of myosin IIA and vinculin. First, myosin IIA is localized considerably more distant from the podosome protrusive modules than vinculin. This was confirmed by fluorescence profile analysis which demonstrated that the highest myosin IIA intensity is detected at ~0.8–0.9 μm from the podosome center, whereas vinculin intensity peaks at ~0.5 μm (Fig. 3b). Second, whereas vinculin occupies the more ventral part of the podosome cluster, myosin IIA is only occasionally found on the ventral side and is mostly detected at a much higher focal plane (Fig. 3a and Supplementary Fig. 4). Quantification demonstrated that the highest fluorescence intensity signal of vinculin is detected at ~50 nm, overlapping with the ventral actin filaments, while myosin IIA intensity peaks well above the ventral network at ~500 nm (470 ± 173 nm) (Fig. 3c). Together, these results support the notion that myosin IIA and vinculin are associated to two different sets of actin filaments.

Since we find myosin IIA at ~500 nm above the ventral plasma membrane (VPM), we hypothesized that at this height, a network of actin filaments must be present. We further reasoned that this network must be very dim and diffraction limited, since we had not seen it before with confocal microscopy. We therefore applied a strong non-linear contrast enhancement (0.3 gamma correction) on the Airyscan actin images taken at the myosin IIA focal plane, and indeed observed a filamentous actin network (Fig. 3d), which we term the dorsal actin filaments. In contrast to the ventral filaments, which only occasionally interconnect neighboring podosomes, the dorsal filaments always span from one podosome to another. Moreover, myosin IIA perfectly colocalizes with these dorsal filaments (Fig. 3d). To further substantiate this finding, we aimed to visualize myosin IIA bipolar filaments to confirm their radial orientation with respect to the podosome core. For this, we simultaneously visualizes the head and tail domains of myosin IIA by staining myosin heavy and light chain and acquired images with Airyscan, which has been exploited

before to visualize myosin IIA filaments in stress fibers[35]. At ~500 nm above the VPM, we observed many myosin IIA bipolar filaments surrounding single podosomes and colocalizing with the dorsal actin filaments (Fig. 3e). Moreover, similar to the dorsal actin filaments, these myosin IIA bipolar filaments are oriented radially with respect to the podosome core (Fig. 3e).

Together, these results demonstrate that the actin filament network radiating from the podosome core is composed of two modules: ventral actin filaments that are associated to vinculin and eventually to integrins, and myosin IIA-crosslinked dorsal actin filaments that may facilitate long-range force transmission between podosomes (Fig. 3f).

**cPM and pPM mostly unaltered on soft vs stiff substrates.** We next aimed to understand how the two protrusion modules and the interpodosomal actin network control podosome mechanosensing. We therefore investigated the podosome nanoscale organization in response to a stiff, non-compliant and a soft, compliant substrate that deforms by podosome protrusive forces. For this, we used two different curing: base ratios of polydimethylsiloxane (PDMS, 1:20 = stiff, ~800kPa; 1:78 = soft, ~1 kPa)[36,37], a polymer that allows cell spreading even at low stiffness[37]. We evaluated the general adhesive capacity of DCs on PDMS, and found that DC spreading and podosome formation was similar on both stiff and soft PDMS (Supplementary Fig. 5a–c). Moreover, similar to what we have shown before on glass[22], podosomes on both stiff and soft PDMS underwent concerted oscillations of actin and vinculin (Supplementary Fig. 6), indicating that general podosome behavior was not altered by substrate stiffness.

To study stiffness-dependent podosome architecture remodeling, we had to ensure that podosome protrusive forces could deform the soft PDMS. We therefore visualized the cell membrane with a fluorescent probe and reasoned that a potential indentation in the soft PDMS due to podosome protrusion should lead to an accumulation of fluorescence intensity around the core due to membrane folding. Indeed, on soft, but not on stiff PDMS, we observed a small but very clear increase in membrane fluorescence intensity directly around the podosome core (Supplementary Fig. 7a). Transmission electron microscopy of transverse sections of cells on stiff and soft PDMS further confirmed deformation of the soft substrates, as small but clear indentations (80 ± 49 nm) were visible underneath podosomes on soft but not on stiff PDMS (Supplementary Fig. 7b).

First, we determined the organization of the two protrusion modules as a function of substrate stiffness. For this, we visualized WASP and α-actinin together with total actin on stiff and soft substrates. On both substrates, we observed a clear localization of WASP to the cPM and α-actinin to the pPM (Fig. 4a–d). Moreover, we observed a differential localization of β and γ-actin to the cPM and pPM, respectively, on both stiff and soft substrates (Supplementary Fig. 8a, b), indicating that the core harbors these two protrusion modules independent of substrate stiffness, and suggests that they are fundamental units for podosome formation.

To determine stiffness-dependent changes in the cPM and pPM architectures, we quantified the fluorescent profiles of actin, WASP, and α-actinin (Fig. 4e–h) as well as the β/γ actin ratio on stiff and soft substrates (Supplementary Fig. 8c, d). We observed a small stiffness-dependent decrease in the FWHM of the actin intensity profile (0.79 ± 0.17 μm on stiff and 0.70 ± 0.18 μm on soft, Fig. 4e), indicating that substrate stiffness slightly affects the size of the protrusive core. No significant differences were observed in the FWHM of WASP (0.48 ± 0.10 μm on stiff and 0.49 ± 0.11 μm on soft, Fig. 4f), indicating that the cPM size is not

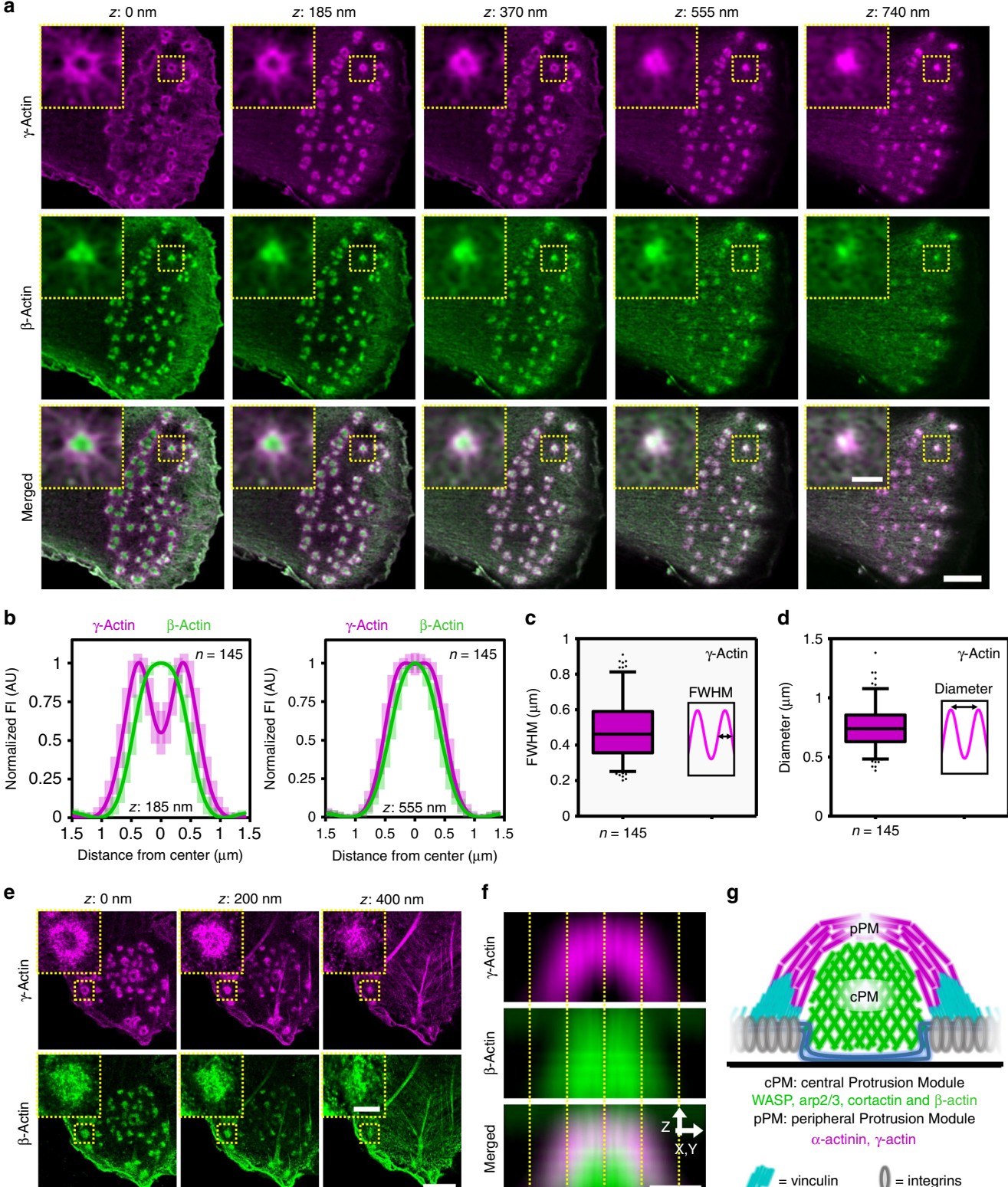

**Fig. 2** γ and β-actin isoforms differentially localize to cPM and pPM. **a** 3D-Airyscan images of a DC stained for γ (magenta) and β-actin (green). Insets depict a single podosome. **b** Average ± s.d. radial fluorescent intensity profile of γ and β-actin ($n = 145$ podosomes pooled from two independent experiments) at two different focal planes ($z$: 185 nm and $z$: 555 nm). **c** Quantification of the FWHM of the γ-actin fluorescent profile ($n = 145$ podosomes pooled from two independent experiments). **d** Quantification of the γ-actin ring diameter ($n = 145$ podosomes pooled from two independent experiments). **e** Dual-color STORM images of a DC stained for γ (magenta) and β-actin (green). Insets depict a single podosome. **f** Average radial orthogonal view of γ (magenta) and β-actin (green) acquired by STORM super-resolution. Bottom panel shows the merged images. **g** Schematic representation of the cPM and pPM in the podosome core. Scale bars: **a** = 2 μm, **e** = 1 μm; **f** = 0.5 μm; insets: **a** = 0.5 μm, **e** = 0.25 μm. FI fluorescent intensity, AU arbitrary units. Box plots indicate median (middle line), 25th, 75th percentile (box) and 5th and 95th percentile (whiskers) as well as outliers (single points). Source data are provided as a Source Data file

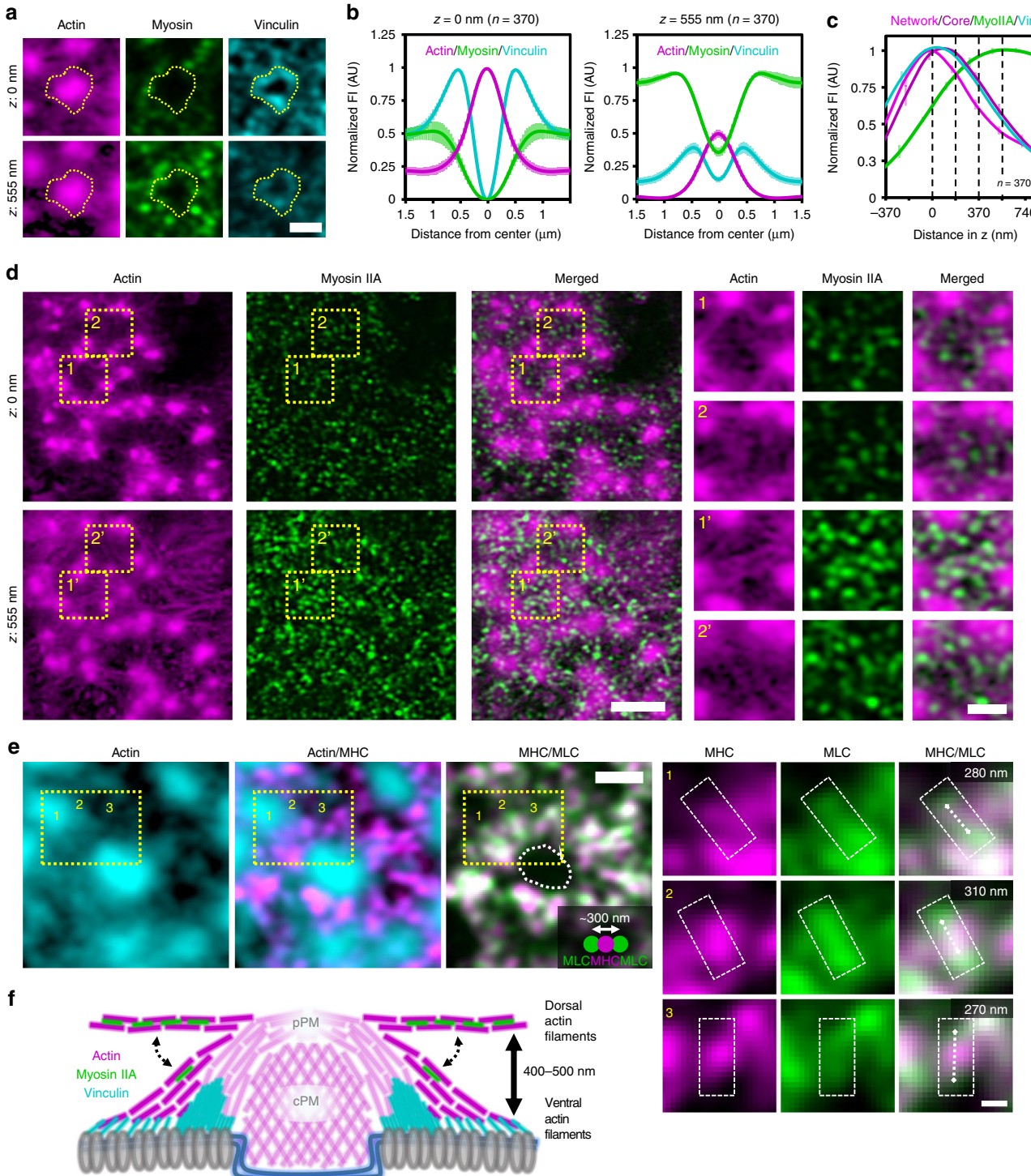

**Fig. 3** Myosin IIA specifically crosslinks dorsal interpodosomal actin filaments. **a** 3D-Airyscan images of a DC stained for actin (magenta), myosin IIA (green), and vinculin (cyan). See Supplementary Fig. 5 for the entire podosome cluster and the additional focal planes. **b** Average ± s.e.m. radial fluorescent intensity profile of actin, vinculin, and myosin IIA (n = 370 podosomes pooled from three independent experiments) at two different focal planes. Data are normalized to all focal planes to emphasize the different intensities of actin and myosin IIA as a function of the focal plane. **c** Quantification of the localization in z of the actin network (light magenta), actin core (dark magenta), vinculin (cyan), and myosin IIA (green) in podosome clusters. The z-sections shown in Supplementary Fig. 5 are represented by the dashed lines in the graph. Shown is the average ± SEM (n = 370 podosomes pooled from three independent experiments). **d** 3D-Airyscan images of a DC stained for actin (magenta) and myosin IIA (green). The contrast of the actin image at 555 nm is strongly enhanced (Gamma correction = 0.3). The zoomed images depict the ventral network (1 and 2) and the dorsal network (1′ and 2′) and associated myosin IIA. **e** Representative Airyscan image of a podosome labelled for actin (cyan), myosin light chain (green), and myosin heavy chain (magenta). The zoomed images depict single myosin IIA bipolar filaments (indicated by dashed rectangle and dashed line) that are oriented radially around the podosome core. The filament length in the upper right corner is the length of the dashed white line. **f** Schematic representation of the localization of vinculin, myosin IIA and the ventral and dorsal actin filaments in podosome clusters. Scale bars: **a** = 1 μm, **d** = 3 μm, **e** = 0.5 μm; zooms: **d** = 1 μm, **e** = 0.1 μm. FI = fluorescent intensity. AU = arbitrary units. Source data are provided as a Source Data file

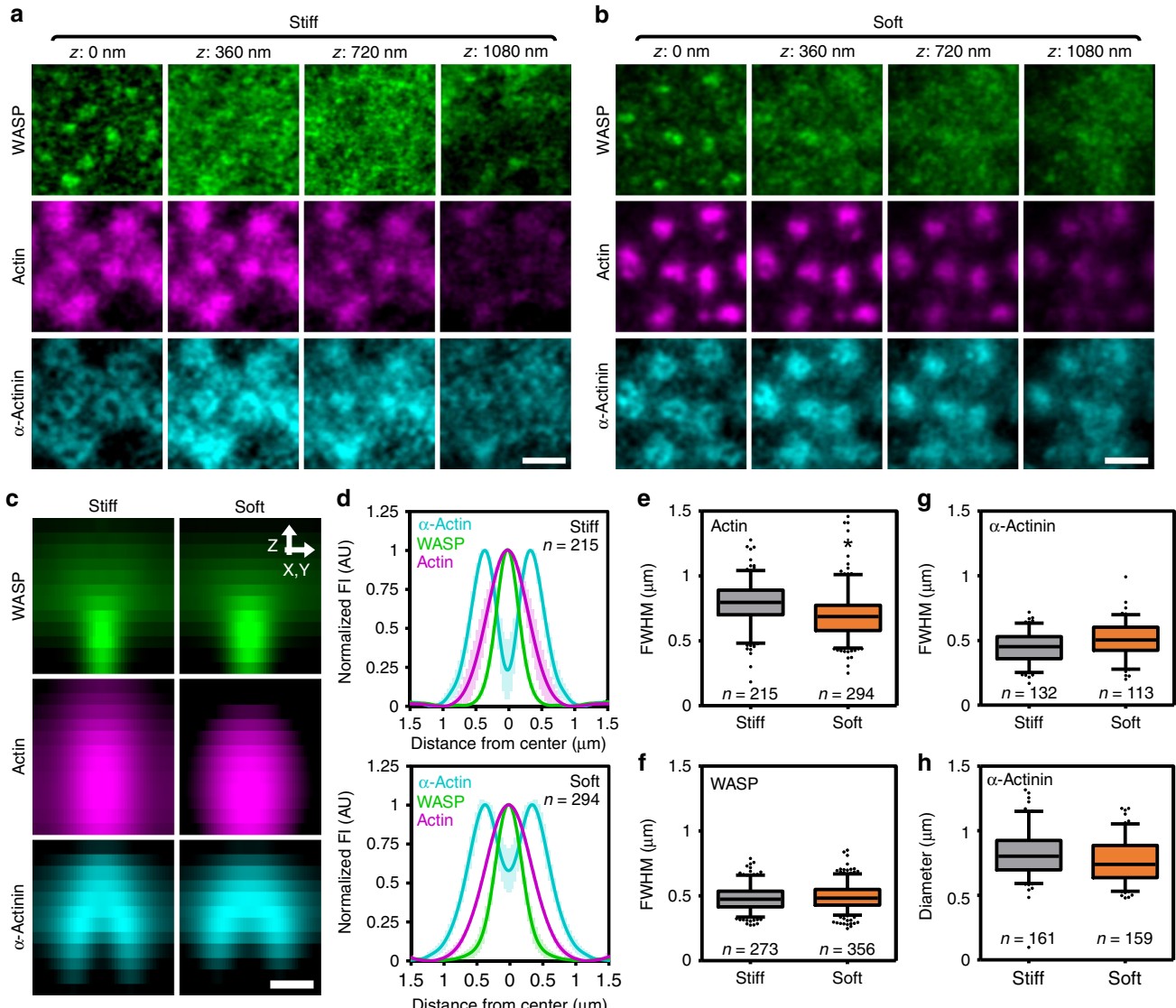

**Fig. 4** cPM and pPM mostly unaltered by changes in substrate stiffness. **a, b** 3D-Airyscan images of DCs transfected with WASP-GFP (green) and α-actinin-HA and stained for HA (cyan) and actin (magenta). Shown are representative images of podosomes on **a** stiff and **b** soft substrate. **c** Average radial orthogonal view of WASP, actin, and α-actinin on stiff ($n = 53$ podosomes) and soft ($n = 45$ podosomes) substrate. **d** Average ± s.e.m. radial fluorescent intensity profile of WASP, actin, and α-actinin on stiff ($n = 113$ podosomes) and soft ($n = 132$ podosomes) substrates. **e** Quantification of the FWHM of the actin fluorescence profiles on stiff ($n = 215$ podosomes) and soft ($n = 294$ podosomes) substrates pooled from three independent experiments. Statistical analysis was performed with an unpaired two-tailed Student's $t$-test. *$P < 0.05$. **f** Quantification of the FWHM of the WASP fluorescence profile on stiff ($n = 273$ podosomes) and soft ($n = 356$ podosomes) substrates. **g** Quantification of the FWHM of the α-actinin fluorescence profiles on stiff ($n = 132$ podosomes) and soft ($n = 113$ podosomes) substrates pooled from three independent experiments. **h** Quantification of the α-actinin ring diameter on stiff ($n = 161$ podosomes) and soft ($n = 159$ podosomes) substrates pooled from three independent experiments. Scale bars: **a, b** = 1 μm, **e** = 0.5 μm. FI fluorescent intensity, AU arbitrary units. Box plots indicate median (middle line), 25th, 75th percentile (box), and 5th and 95th percentile (whiskers) as well as outliers (single points). Source data are provided as a Source Data file

affected by substrate stiffness. For α-actinin, we observed a small, non-significant increase in the FWHM of the fluorescent intensity profile (0.44 ± 0.11 μm on stiff and 0.51 ± 0.13 μm on soft, Fig. 4g), as well as a small, non-significant decrease in the pPM diameter (0.82 ± 0.18 μm on stiff and 0.77 ± 0.17 μm on soft, Fig. 4h), indicating that the pPM is also largely unaffected by changes in substrate stiffness. Lastly, both immunofluorescence analysis of Airyscan images and western blot analysis of VPMs demonstrated no differences in the β/γ actin ratio as a function of substrate stiffness (Supplementary Fig. 8c, d), supporting the notion that the cPM and pPM architecture is not affected by substrate stiffness.

**Ventral filaments reorganize in response to soft substrates.**
Next, we investigated the organization of the dorsal and ventral actin filaments as a function of substrate stiffness. For the dorsal network, we determined the localization and activation of myosin IIA. First, we observed no difference in the amount of myosin IIA at podosomes on stiff and soft substrates (Fig. 5a–c). Second, the lateral organization of myosin IIA appeared unaffected by changes in substrate stiffness with myosin IIA peak intensity at ~0.8–0.9 μm from the podosome core center (Fig. 5d). Third, myosin IIA was located ~500 nm (523 ± 162 nm on stiff and 490 ± 190 nm on soft) above the ventral actin network on both stiff and soft substrates (Fig. 5e, Supplementary Fig. 9). Lastly, to

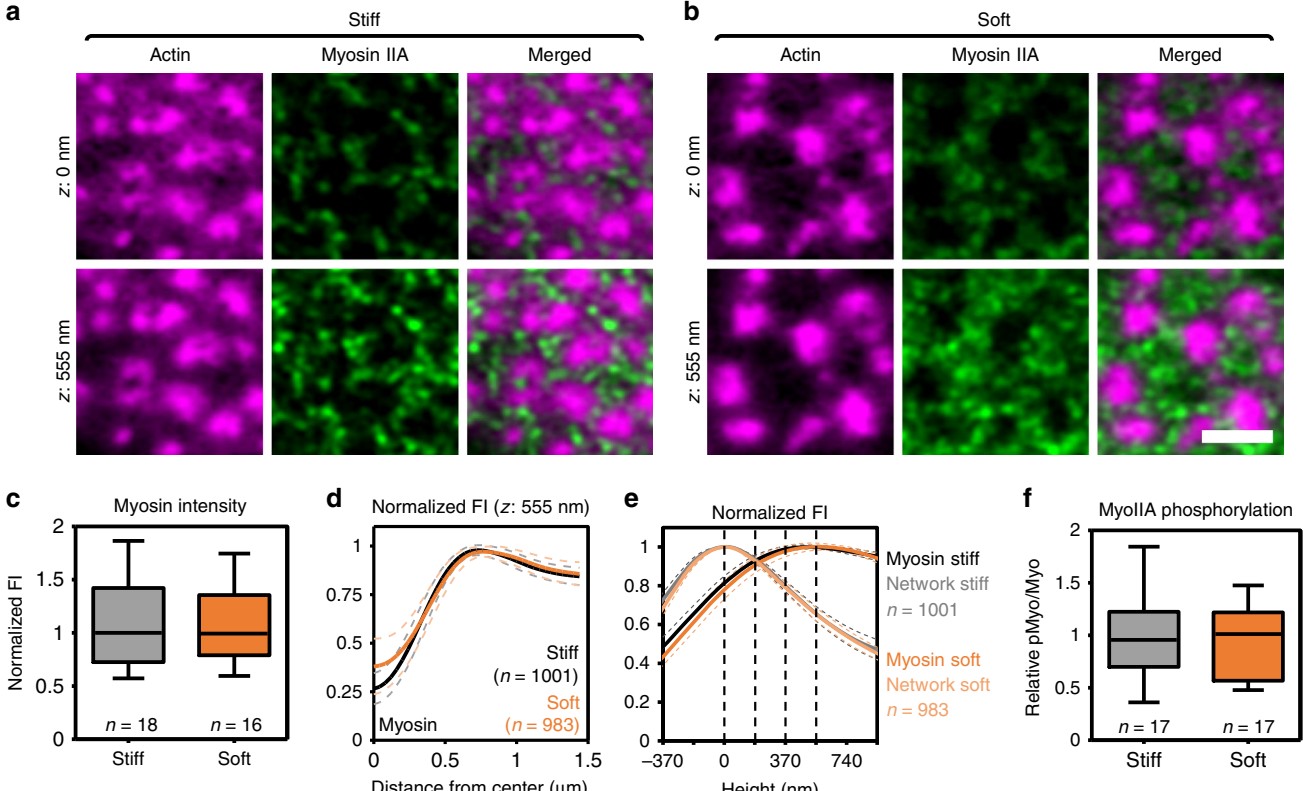

**Fig. 5** Myosin IIA localization and activation unaffected by changes in substrate stiffness. **a**, **b** 3D-Airyscan images of DCs stained for actin (magenta) and myosin IIA (green). Shown are representative images of podosomes on **a** stiff and **b** soft substrate. **c** Quantification of the intensity of the myosin IIA signal in podosome clusters on stiff ($n = 18$ clusters) and soft ($n = 16$ clusters) substrates pooled from three independent experiments. **d** Average ± s.e.m. radial orthogonal view of myosin on stiff ($n = 1001$ podosomes) and soft ($n = 983$ podosomes) substrates pooled from three independent experiments.
**e** Quantification of the localization in z of the actin network (light colors) and myosin IIA (dark colors) in podosome clusters on stiff ($n = 1001$ podosomes) and soft ($n = 983$ podosomes) substrates pooled from three independent experiments. The dashed lines in the graph represent the z-sections, two of which are shown in **a**. **f** DCs were seeded on soft and stiff substrates and stained for myosin IIA and phospho-myosin light chain. The graph depicts the quantification of the pMyo/Myo ratio on stiff ($n = 17$ podosome clusters) and soft ($n = 17$ podosome clusters) substrates pooled from three independent experiments. Scale bar = 2 μm. FI fluorescent intensity, AU arbitrary units. Box plots indicate median (middle line), 25th, 75th percentile (box) and 5th and 95th percentile (whiskers) as well as outliers (single points). Source data are provided as a Source Data file

determine the activation status of myosin IIA, we analyzed myosin light chain phosphorylation by immunofluorescence microscopy and did not observe any differences between stiff and soft substrates (Fig. 5f). Together, these data indicate that myosin IIA localization and activation at podosome clusters are unaffected by substrate stiffness and strongly suggest that the dorsal actin filaments are not the primary players in podosome stiffness sensing.

Next, we analyzed the ventral actin filaments by super-resolution microscopy. Interestingly, we found a significant decrease in the length of these filaments on soft substrates (0.43 ± 0.13 μm on stiff vs. 0.26 ± 0.11 μm on soft) (Fig. 6a, also visible in Fig. 5a, b). Since, within the podosome cluster, the ventral actin filaments direct the localization of tension-sensitive proteins vinculin and zyxin but not of the scaffold protein talin[17,22], we characterized the localization of vinculin, zyxin, and talin in response to changes in substrate stiffness. For vinculin on stiff substrates, we observed a localization close to the podosome core as well as in areas in between the cores (Fig. 6b), similar to what we had reported before on glass[22]. Remarkably, on soft substrates, while the levels of vinculin did not change (Supplementary Fig. 10a), a reorganization occurred whereby vinculin appeared much more confined to the core (Fig. 6b), something which we confirmed in living cells transfected with Lifeact-GFP and vinculin-mCherry (Fig. 6c, Supplementary Fig. 10b). This

resulted in a significant decrease in both the width (0.71 ± 0.22 μm on stiff vs. 0.61 ± 0.20 μm on soft) and the diameter (1.02 ± 0.23 μm on stiff vs. 0.92 ± 0.23 μm on soft) of the vinculin ring (Fig. 6d, e). Importantly, we observed an analogous reorganization for zyxin (Fig. 6f), but not for talin (Fig. 6g), suggesting that this stiffness-dependent response is specific for proteins for which their positioning is known to be controlled by the ventral filaments. In this regard, it is also interesting to note that on all of the substrates, the vinculin pool that was more distant from the core colocalized with the ventral actin filaments (Fig. 6h). Importantly, neither inhibition nor activation of myosin IIA affected the localization of vinculin on stiff and soft substrates (Supplementary Figs. 11 and 12). This further confirms the existence of two actin networks and demonstrates that substrate stiffness selectively induces a nanoscale reorganization of the ventral actin filaments and their associated mechanosensory proteins, strongly suggesting that these filaments, and not the protrusion modules or the dorsal actin filaments, are the primary mechanosensors in podosome clusters.

**Stiffness controls podosome connectivity and degradation.** We have recently demonstrated that the interpodosomal actin filaments facilitate podosome mesoscale connectivity that plays a role in the generation of dynamic spatial patterns of podosome

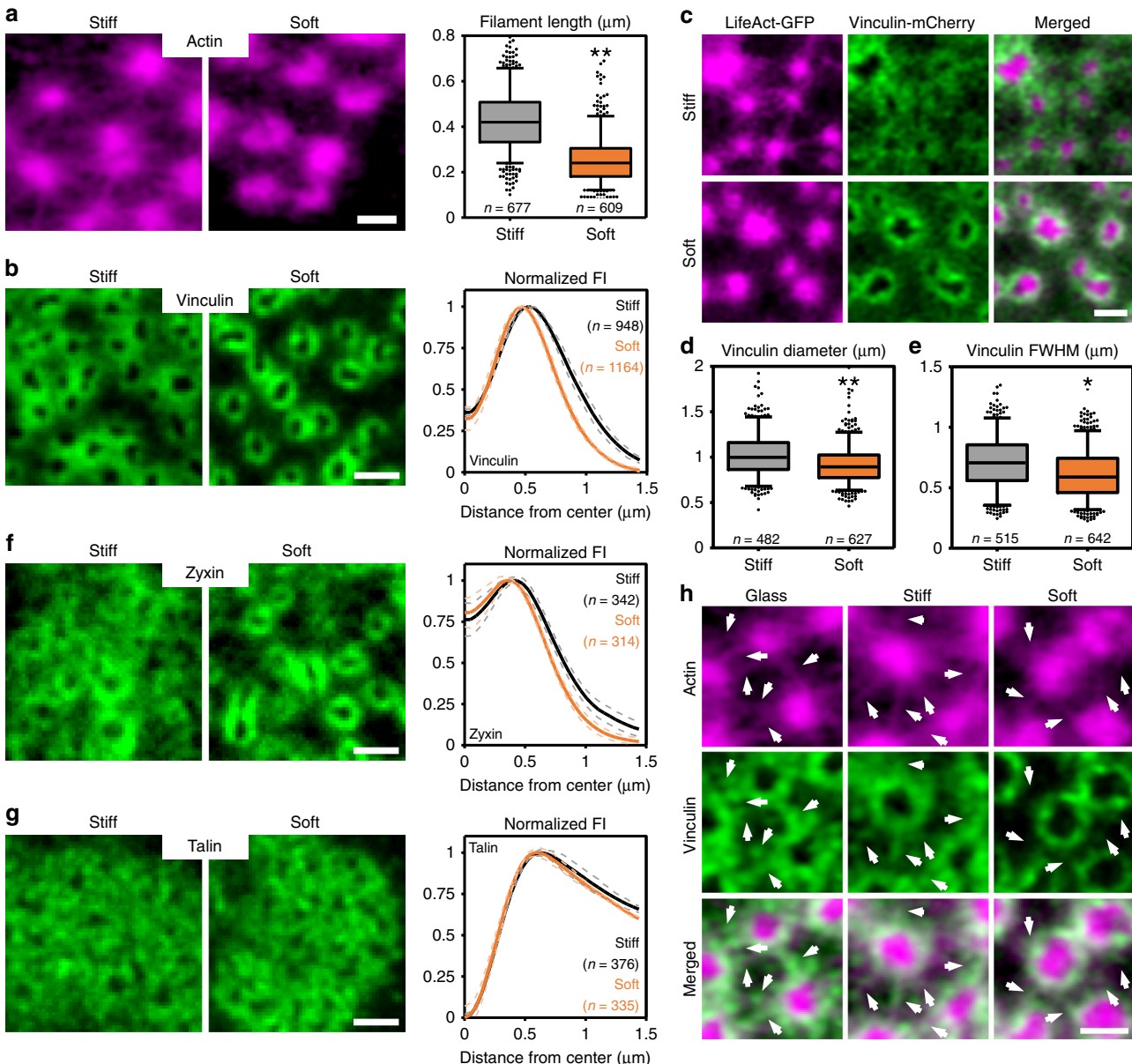

**Fig. 6** Reorganization of ventral actin filaments in response to soft substrates. **a** Airyscan images of DCs stained for actin. Shown are representative images of podosomes on stiff (left) and soft (right) substrates. The graph depicts the quantification of the length of the radiating actin filaments on stiff ($n = 677$ filaments) and soft ($n = 609$) substrates pooled from three independent experiments. Statistical analysis was performed with an unpaired two-tailed Student's $t$-test. **$P < 0.01$. **b** Widefield images of DCs stained for vinculin. Shown are representative images on stiff (left) and soft (right) substrate. The graph depicts the average ± s.e.m. radial fluorescence intensity profile for vinculin on stiff ($n = 948$ podosomes) and soft ($n = 1164$) substrates pooled from three independent experiments. **c** Airyscan images of DCs transfected with Lifeact-GFP (magenta) and Vinculin-mCherry (green) and seeded on stiff and soft substrates. Complete cluster is shown in Supplementary Fig. 11. **d** Quantification of the vinculin ring diameter on stiff ($n = 482$) and soft ($n = 627$) substrates pooled from three independent experiments. Statistical analysis was performed with an unpaired two-tailed Student's $t$-test. **$P < 0.01$ **e** Quantification of the FWHM of the vinculin ring on stiff ($n = 515$ podosomes) and soft ($n = 642$ podosomes) substrates. Statistical analysis was performed with an unpaired two-tailed Student's $t$-test pooled from three independent experiments. *$P < 0.05$. **f** Widefield images of DCs stained for zyxin. Shown are representative images on stiff (left) and soft (right) substrate. The graph depicts the average ± s.e.m. radial fluorescence intensity profile for zyxin on stiff ($n = 342$ podosomes) and soft ($n = 314$) substrates) substrates pooled from two independent experiments. **g** Widefield images of DCs stained for talin. Shown are representative images on stiff (left) and soft (right) substrate. The graph depicts the average ± s.e.m. radial fluorescence intensity profile for talin on stiff ($n = 376$ podosomes) and soft ($n = 335$) substrates pooled from two independent experiments. **h** Airyscan images of DCs stained for actin (magenta) and vinculin (green). Shown are representative images of podosomes on glass (left), stiff (middle), and soft (right) substrate. Arrows indicate the location of the radiating actin filaments and associated vinculin. Scale bars: **a**–**c** = 1 μm, **f**, **g** = 1 μm, **h** = 0.5 μm. FI fluorescent intensity, AU arbitrary units. Box plots indicate median (middle line), 25th, 75th percentile (box) and 5th and 95th percentile (whiskers) as well as outliers (single points). A non-linear contrast enhancement (gamma correction = 0.5) was applied to all actin images (in **a**, **c**, and **h**) to better visualize the ventral filaments. Source data are provided as a Source Data file

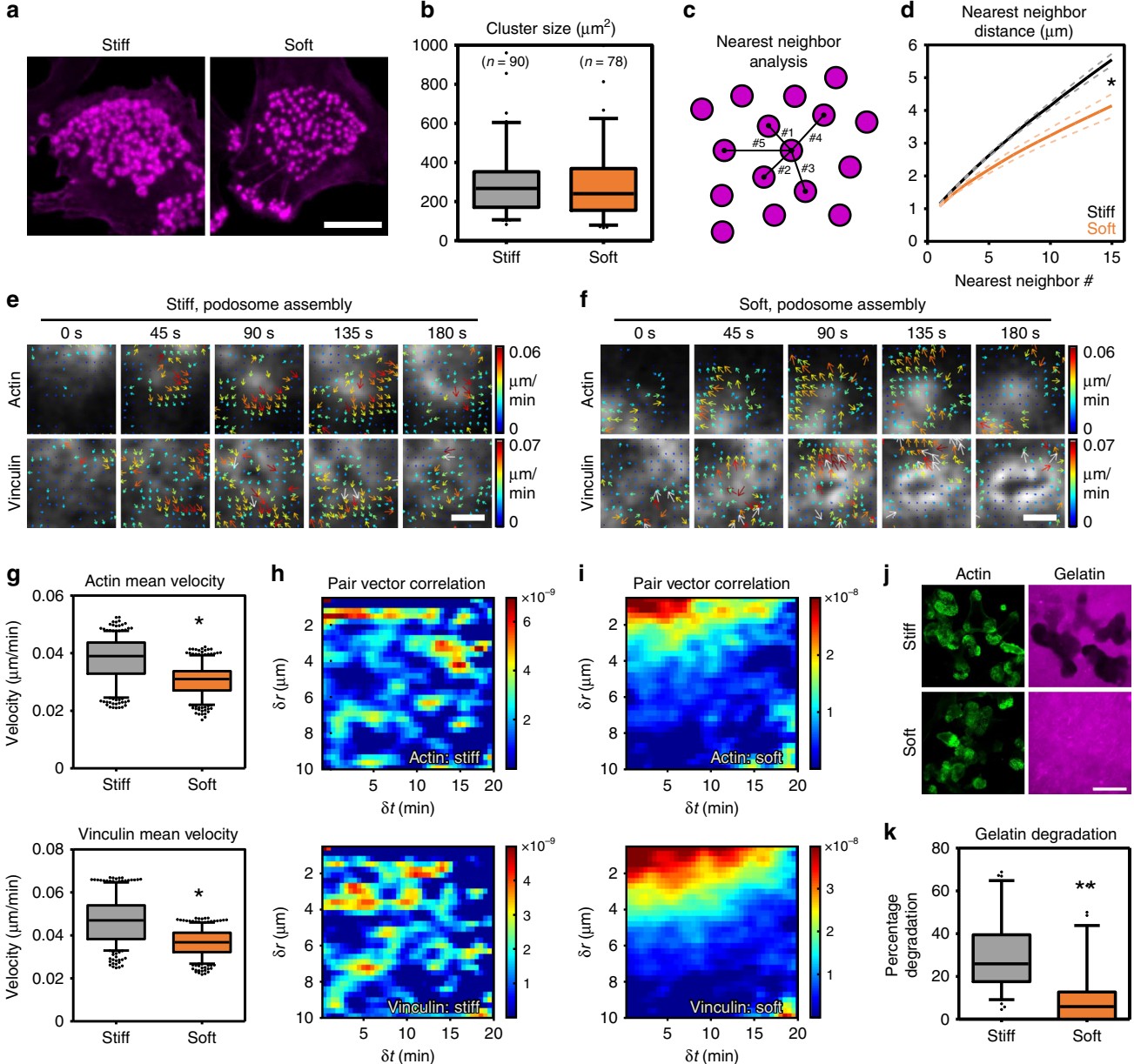

**Fig. 7** Substrate stiffness controls podosome mesoscale connectivity and degradative capacity. **a** Widefield images of DCs stained for actin. Shown are representative podosome clusters on stiff (left) and soft (right) substrates. **b** Quantification of the podosome cluster size on stiff ($n = 90$ clusters) and soft ($n = 78$ clusters) substrates pooled from three independent experiments. **c** Graphical explanation of the nearest-neighbor analysis. **d** Quantification of the nearest-neighbor distance for podosomes in clusters of at least 15 podosomes on stiff ($n = 1652$ podosomes) and soft ($n = 1470$ podosomes) substrates pooled from three independent experiments. Statistical analysis was performed with an unpaired two-tailed Student's t-test. *$P < 0.05$. **e**, **f** DCs were transfected with Lifeact-GFP and vinculin-mCherry. Imaging was performed using Airyscan confocal microscopy with 15 s frame intervals. Time series were subjected to twSTICS analysis and results are plotted as vector maps in which the arrows indicate direction of flow and both the size and color coding are representative of the flow magnitude. Shown are representative waves of vectors for actin and vinculin on **e** stiff and **f** soft substrates. **g** Quantification of the mean velocity of actin (upper panel) and vinculin (lower panel) on stiff and soft substrates using STICS. Statistical analysis was performed with an unpaired two-tailed Student's t-test ($n = 5$ cells pooled from three independent experiments). *$P < 0.05$. **h, i** Pair vector correlation analysis for actin and vinculin on **h** stiff and **i** soft substrates that indicate the spatial and temporal scales of vector correlation of the twSTICS analysis. Shown are the average pair vector correlations from five time series. **j** DCs were seeded on gelatin-rhodamine (magenta)-coated stiff and soft substrates, incubated overnight and subsequently stained for actin (green). Shown are representative images of gelatin degradation of stiff (upper panels) and soft (lower panels) substrates. **k** Quantification of the degraded area on both stiff and soft substrates. Statistical analysis was performed with an unpaired two-tailed Student's t-test ($N = 6$ independent experiments). **$P < 0.01$. Scale bars: **a** = 10 μm, **e** = 0.5 μm, **f** = 0.5 μm, **j** = 20 μm. Box plots indicate median (middle line), 25th, 75th percentile (box) and 5th and 95th percentile (whiskers) as well as outliers (single points). Source data are provided as a Source Data file

cytoskeletal components[18]. We therefore investigate whether the mesoscale connectivity was altered in response to substrate stiffness. We first determined podosome cluster area and found no difference between clusters assembled on stiff or soft

substrates (Fig. 7a, b). Next, we determined the local podosome density as calculated by the nearest-neighbor distance (NND) between podosomes in clusters containing at least 15 podosomes (Fig. 7c, d). Interestingly, the NND was significantly smaller on

the soft compared to stiff substrates (Fig. 7d). Podosomes are thus capable of organizing in higher ordered clusters independent of substrate stiffness, but substrate stiffness does affect local podosome density.

To determine whether substrate stiffness affects the mesoscale connectivity of the podosome clusters, we used our recently developed sliding time window spatiotemporal image correlation spectroscopy (twSTICS)[18] to analyze the Lifeact-GFP/vinculin-mCherry movies obtained by Airyscan live cell imaging. By generating time-evolving vector maps, twSTICS maps the velocity (magnitude and direction) of flowing fluorescent biomolecules imaged within the cell and can therefore be used to quantify the properties of dynamic cellular features. twSTICS revealed many coordinated flows of actin and vinculin within the podosome cluster on both the stiff and soft substrates (Fig. 7e, f, Supplementary Figs. 13 and 14, Supplementary Movies 4 and 5), indicating that actin and vinculin are dynamic and display correlated movement independent of substrate stiffness. Since regions with relatively stable podosomes or without podosomes did not produce any measurable flows, we conclude that the flows detected by the twSTICS analysis must originate from podosomes that undergo vertical oscillations (Supplementary Figs. 13 and 14, Supplementary Movies 4 and 5).

From the twSTICS measured flows, we first calculated the mean velocity and found a consistently lower mean velocity for both actin (0.05 ± 0.01 μm/min on stiff vs. 0.04 ± 0.01 μm/min on soft) and vinculin (0.06 ± 0.01 μm/min on stiff vs. 0.05 ± 0.01 μm/min on soft) on soft compared to stiff substrates (Fig. 7g). We next investigated the directionality of the actin and vinculin fluxes by performing pair vector correlation (PVC) analysis on the twSTICS-generated vector maps. For this, we calculated a vector dot product correlation function over all vector pairs as a function of their spatial separations and time differences[18]. For both actin and vinculin, we observed a striking difference in the PVC distribution between the stiff and soft substrates. On stiff PDMS, we found small clusters of correlated vectors that were regularly organized in space (up to 10 μm) and time (up to 20 min) (Fig. 7h), indicating that podosomes oscillate at a steady periodicity throughout the cluster over relatively long periods of time. On soft substrates, however, we found that the vectors are very strongly correlated only over short distances (2–3 μm) but with a clear periodicity over long periods of time (up to 20 min) (Fig. 7i). Thus, on softer substrates, the actin and vinculin flows due to vertical oscillations are only locally correlated in space and do not span the entire cluster, suggesting that podosome mesoscale connectivity is specifically increased when the cell must respond to stiff, non-deformable, substrates.

Podosomes have the ability to degrade extracellular matrix, presumably to create weak spots that become permissive for deformation and protrusion. Since podosomes exhibit a different collective behavior when exposed to a deformable substrate, we postulated a concomitant decrease in their degradative capacity. To test this, we seeded cells on stiff and soft substrates that have been previously coated with rhodamine-labelled gelatin. On stiff substrates, the gelatin coating was readily degraded (Fig. 7j, k, Supplementary Fig. 15), with degradation clearly occurring underneath the podosome clusters as observed in living DCs (Supplementary Fig. 15a, Supplementary Movie 6). On the contrary, we observed a strong and significant decrease in the capability of podosomes to degrade gelatin on soft substrates (Fig. 7j, k, Supplementary Fig. 15b). This indicates that substrate deformation controls the degradative function of podosomes and demonstrates that podosome mesoscale connectivity and their ability to degrade extracellular matrix are functionally connected.

## Discussion

In this study, we unraveled the modular architecture of actin that enables protrusion and mechanosensing by podosomes. By combining super-resolution microscopy and extensive quantitative image analysis we reveal that the podosome protrusive core consists of a cPM encased by a pPM, each module harboring specific actin interactors and actin isoforms. Also, we show that from the core, two actin modules radiate: ventral filaments bound by vinculin and connected to the plasma membrane and dorsal interpodosomal filaments crosslinked by myosin IIA. We further demonstrate that on stiff substrates, the actin modules mediate long-range substrate exploration, associated with degradative behavior. On protrusion-permissive substrates, where less tension is exerted, the vinculin-bound ventral actin module shortens, resulting in short-range connectivity and a focally protrusive, non-degradative state. Our findings redefine podosome nanoscale architecture, demonstrating how actin modularity enables invadosome mechanosensing in cells that breach tissue boundaries.

Many actin-associated proteins such as WASP, arp2/3, cortactin, and α-actinin have been identified in the podosome core but their exact nanoscale positioning remained elusive. WASP and cortactin were found previously to locate to the base of podosome cores in osteoclasts[38], while the localization of α-actinin has been more promiscuous with some reports suggesting colocalization with the actin core and others colocalization with the vinculin ring[39–42]. Using super-resolution imaging and detailed image analysis, we here provide a modular framework for the podosome protrusive apparatus that blurs the traditional core-ring concept (Supplementary Fig. 16). Branched actin-associated proteins such as WASP, Arp2/3, and cortactin locate primarily to the center of the core, i.e. the cPM, while linear actin-associated proteins, such as α-actinin are primarily found in the core periphery, i.e. the pPM. Interestingly, at the ventral side of podosomes, α-actinin partially colocalizes with vinculin, indicating that also vinculin is bound to part of the pPM, presumably providing linkage of the pPM to the plasma membrane. Our findings explain previous observations that the actin core of individual podosomes is dome-shaped[43]. Moreover, we propose here that the pPM includes the podosome cap, a substructure that has been identified previously based on the localization of the formin INF2, supervillin, and LSP-1 at the top of the core[44–46]. Like α-actinin, INF2 and supervillin are primarily associated with linear actin filaments supporting the notion that the pPM consists of this type of filaments. It remains to be determined where the pool of pPM actin is polymerized but the presence of formins at the top of podosome cores suggests this as the likely site of pPM actin polymerization.

The modularity of the podosome core is further substantiated by our finding that actin isoforms are spatially segregated at podosomes with the cPM enriched in β-actin and the pPM in γ-actin. Understanding the non-redundant function of actin isoforms in cytoskeletal remodeling is an area of steeply emerging interest[47–49]. Although not much is known about the different functions of the two isoforms, it is generally assumed that β-actin is preferentially located to protrusive lamellipodial structures and γ-actin to stress fibers[31]. Interestingly, it is well known that lamellipodial structures mainly contain branched actin while stress fibers are usually composed of linear actin filaments[50]. The differential localization of β and γ-actin observed in the podosome core therefore strongly supports the notion that the core consists of two structurally and functionally distinct actin-based modules; a cPM of branched actin and a pPM of linear actin. It remains unclear how the differential integration of β and γ-actin monomeric subunits is spatially regulated. To our knowledge, no reports exist that describe the preferential association of actin-interacting proteins with a particular actin isoform. Considering

our results, one would argue that WASP/arp2/3-mediated actin polymerization mainly incorporates β-actin and formin-mediated nucleation preferentially incorporates γ-actin. Another possible explanation for the differential organization of actin isoforms could be the directed localization of actin mRNA. Local actin translation has been proposed to control β-actin enrichment in lamellipodia[51] and considering the abundance of ribosomal proteins at podosomes[25], it is tempting to speculate that a portion of β-actin is locally translated to generate the cPM while γ-actin is recruited from the cytoplasm to generate the pPM and the interpodosomal filaments.

It is generally accepted that podosome-mediated protrusive force generation is regulated by an interplay between actin polymerization and myosin IIA activity[21,22,43]. Furthermore, recent in silico modelling of podosome force distribution strongly suggested that protrusive force generation in the core is balanced by local pulling force in the ring at the level of single podosomes[20]. An explanation, however, for how protrusive and pulling forces are transmitted within the podosome structure remained elusive, since no clear structural connection between the core and the ring has been described so far. Based on our results, we now propose that the classical core-ring model inadequately explains podosome force generation, and present a fully integrated structure–function model for how protrusion and mechanosensing may be regulated by podosomes (Supplementary Fig. 16). In this model, podosome protrusive forces are intrinsically balanced by the modular architecture of individual podosomes. cPM actin polymerization generates a downward protrusive force that is initially balanced by an upward counterforce from the underlying substrate. The subsequent vertical growth of the cPM actin generates an upward force at the top of podosomes that is counterbalanced by the pPM actin encasing the entire cPM. We hypothesize that there is a direct association of the pPM to adaptor proteins, such as vinculin to provide adhesion and mechanical stability, and thereby blurring the classical concepts of a podosome core and ring. Interestingly, a recent report by Revach et al. suggested that invadopodia, which are only occasionally surrounded by vinculin, are mechanically stabilized by the nucleus[52]. Yet, the same report also showed that the loss of mechanical support from the nucleus is rescued by the recruitment of vinculin to invadopodia[52]. In DCs, podosome clusters are rarely located underneath the nucleus, indicating that a mechanical interplay between the nucleus and podosomes is unlikely and support from vinculin-based adhesion is therefore always required to provide mechanical stability. Overall, we propose that all invadosomes require mechanical stability for protrusion and that the modules that provide this stability are universal and can be adapted depending on local cellular circumstances.

We show that two interpodosomal networks exist: a ventral network that is associated to vinculin and a dorsal network that is crosslinked by myosin IIA (Supplementary Fig. 16). Our previous work has shown that the interpodosomal actin filaments are important for interconnecting neighboring podosomes[18,19], but we now demonstrate that it is primarily the dorsal network that interconnects neighboring podosomes, while the ventral network acts as the primary mechanosensing element in podosomes. In contrast to focal adhesions[53], podosomes have the ability to assemble under conditions with low or no traction forces[33,54,55]. The detailed organization, however, of podosomes on substrates with different stiffness had not been studied so far. We now find that the ventral actin filaments shorten on compliant substrates, where less tension and more protrusion are exerted. The role of the ventral actin filaments in mechanosensing is supported by previous findings that these filaments are associated with mechanosensitive

proteins such as vinculin[17,19,22]. Interestingly, shortening of the ventral actin filaments is accompanied by enhanced local clustering of podosomes, as well as a decreased mesoscale connectivity on soft substrates. Since podosome mesoscale connectivity is thought to facilitate basement membrane exploration for protrusion-permissive spots, our data clearly suggest that substrate stiffness or deformability provides feedback for the clusters while exploring their surroundings.

How do local cell-substrate traction forces control the length of the ventral actin filaments? Our results indicate that myosin IIA does not contribute to the mechanical response of podosomes. A possible mechanism for podosome mechanosensing is that altered actin polymerization kinetics locally within the cPM directly control the mechanical response of podosomes. Arp2/3-mediated actin polymerization has been shown to be dependent on mechanical stimuli both in reconstitution assays[56,57] as well as in living cells[58]. Altered polymerization kinetics of the cPM in response to compliant substrates could therefore very well result in local changes in the G- to F-actin ratio, which has been shown to control formin-dependent actin polymerization[59] and which could therefore eventually lead to a reorganization of the ventral actin filaments.

We find that podosome-mediated matrix degradation is regulated by the physical properties of the microenvironment. Interestingly, stiffness-dependent degradation has been shown before in invadopodia[60,61] suggesting that matrix degradation by podosomes and invadopodia is controlled by similar mechanisms. Alexander et al. showed that the increased matrix degradation of invadopodia on stiff substrates was regulated through a myosin IIA-dependent pathway[60]. Furthermore, for macrophage podosomes, it has been shown that knockdown of myosin IIA results in a reduction of matrix degradation[44] and that the absence of cell-substrate traction forces results in the absence of MT1-MMP[54]. Although we cannot exclude that myosin IIA also plays a role in the stiffness-dependent decrease in degrading activity of podosomes in DCS, we did not observe any change in myosin IIA localization or activity as a function of substrate stiffness. Yet, it may still be that myosin IIA dynamics is altered in podosome clusters on soft substrates such that the trafficking and fusion of MT1-MMP-positive vesicles are impaired. Another explanation is that other tension sensitive adaptor proteins regulate the activity or excretion of metalloproteases due to altered force distributions in podosome modules on soft substrates. We recently showed that on stiff substrates, MT1-MMP-dependent gelatin degradation is mediated by the action of phospholipase D[62]. It would be interesting to explore the role of this signaling pathway in podosome mechanosensing in future studies. Finally, substrate degradation is also observed at focal adhesions, in agreement with previous work[63], and it would be interesting in the future to compare the effect of substrate stiffness on extracellular matrix degradation by different adhesive structures.

In conclusion, our results indicate that protrusion and mechanosensing is controlled by the modular architecture of podosomes. Protrusion is controlled by two cooperating core modules and podosomes respond to lower substrate stiffness by reorganizing their ventral radiating filaments and associated proteins, thereby enhancing local clustering, changing their dynamic behavior and decreasing their degradative capacity. Podosomes thus functionally adapt from an explorative, degradative behavior on stiff substrates to a focally protrusive, non-degradative state on soft substrates. Our findings highlight how stiffness-induced nanoscale architectural changes can control the mesoscale collective behavior of protrusive podosomes and reveal how actin-based cytoskeletal structures allow cells to breach tissue boundaries and basement membranes.

## Methods

**Generation of human DCs.** DCs were generated from peripheral blood mononuclear cells (PBMCs)[64,65]. Monocytes were derived either from buffy coats or from a leukapheresis product, purchased at Sanquin blood bank, Nijmegen, the Netherlands. PBMCs were isolated by Ficoll density gradient centrifugation (GE Healthcare Biosciences, 30 min, 4 °C, 2100 r.p.m.). PBMCs were extensively washed in cold phosphate buffered saline (PBS) supplemented with 0.1% (w/v) bovine serum albumin (BSA, Roche Diagnostics) and 0.45% (w/v) sodium citrate (Sigma Aldrich). PBMCs were seeded in plastic culture flasks for 1 h and monocytes were isolated by plastic adherence. Monocytes were cultured in RPMI 1640 medium (Life Technologies) supplemented with fetal bovine serum (FBS, Greiner Bio-one), 1 mM ultra-glutamine (BioWhittaker), antibiotics (100 U ml$^{-1}$ penicillin, 100 µg ml$^{-1}$ streptomycin, and 0.25 µg ml$^{-1}$ amphotericin B, Gibco) for 6 days, in a humidified, 5% $CO_2$-containing atmosphere. During these 6 days DC differentiation was induced by addition of IL-4 (500 U ml$^{-1}$) and GM-CSF (800 U ml$^{-1}$) to the culture medium. At day 5 or day 6 cells were collected and reseeded onto coverslips or imaging dishes.

**Generation of murine BMDCs.** DCs were generated from murine bone marrow isolated from the femur/tibia of mice. The mice were bred and housed at the Animal Research Facility of the Radboud University Medical Center. All animal experiments were documented and approved by the Animal Experimental Committee of the Radboud University Medical Center and were performed in accordance with regulatory standards of the Animal Experimental Committee. Batf3-dependent CD103 + DCs (CD11cposB220negCD103pos)[66] were generated by culturing bone marrow cells in RPMI 1640 supplemented with 10% FCS, 0.5% antibiotic–antimycotic, 1% ultra-glutamine, 50 µM β-mercaptoethonal, 5 ng/ml mGM-CSF, and 200 ng/ml human rFlt3L, fresh medium was added at day 6 and cells were replated in fresh medium at day 9. Cells were harvested and used for experiments at day 14.

**Antibodies and reagents.** The following primary antibodies were used (dilution is indicated for immunofluorescence unless stated otherwise): anti-α-actinin (ab18061, Abcam, 1:100 dilution), anti-HA (3F10, Sigma-Aldrich, 1:200 dilution), anti-vinculin (#V9131, Sigma-Aldrich, 1:400 dilution), anti-zyxin (sc-6437, Santa Cruz, 1:40 dilution), anti-talin (#T3287, Sigma-Aldrich, 1:100 dilution), anti-β-actin (#MCA5775GA, Bio-Rad Laboratories, 1:200 dilution, 1:2000 for western blot), anti-γ-actin (#MCA5776GA, Bio-Rad Laboratories, 1:200 dilution, 1:2000 for western blot), anti-actin (#A2066, Sigma-Aldrich, 1:5000 dilution for western blot), anti-myosin IIA (#909802, BioLegend, 1:100 dilution), and anti-phospho-myosin light chain (#3671, Cell Signalling Technology, 1:100 dilution). Secondary antibodies conjugated to Alexa647, Alexa555, or Alexa568 were used (Life Technologies, 1:400 dilution). Actin was stained with Alexa488/Alexa633-conjugated phalloidin (Life Technologies, 1:200 dilution). Blebbistatin was purchased from Sigma-Aldrich (#B0560, 50 µM for 60 min).

**VPM preparation.** To prepare VPMs, cells on PDMS-coated Willco Wells were briefly sonicated. Sonication was performed using a Sartorius Labsonic P sonicator with cycle set at 1 and amplitude at 20% output. The sonicator tip was placed in a glass beaker containing 100 ml prewarmed hypotonic PHEM buffer (20% PHEM: 6 mM PIPES, 5 mM HEPES, 0.4 mM $Mg_2SO_4$, 2 mM EGTA). Willco Wells were held 1–2 cm below the sonicator tip at a 45° angle in the hypotonic PHEM solution. Cells were sonicated for ~3 s and directly after sonication, the remaining VPMs were solubilized in 1% SDS lysis buffer for analysis of β and γ-actin using western blot.

**Constructs.** Lifeact-BFP and Lifeact-iRFP were generated by replacing GFP in the Lifeact-GFP construct (gift from Michael Sixt) for BFP and iRFP. For BFP, tagBFP-N1 construct (Invitrogen) was digested with AgeI and NotI and BFP was ligated into Lifeact-GFP construct. For iRFP, a PCR product was generated with forward primer 5′-ATCGACCGGTCGCCACCATGGCTGAAGGATCCGTCG-3′ and reverse primer 5′-CGATGCGGCCGCTCACTCTTCCATCACGCCGAT-3′ using the iRFP-PH-PLCδ construct as template (gift from Pietro De Camilli). The PCR product was subsequently digested with AgeI and NotI and ligated into Lifeact-GFP vector. α-actinin-tagRFP was generated by replacing mEOS3.2 from α-actinin-mEOS3.2 (Addgene 57444) with the tagRFP sequence from ptagRFP-N1 using AgeI and NotI restriction sites. α-actinin-HA was generated by annealing the forward 5′-CCGGTCGCCACCTACCCATACGATGTTCCAGATTACGCTTGA GC-3′ and reverse oligo 5′-GGCCGCTCAAGCGTAATCTGGAACATCGTATGG GTAGGTGGCGA-3′ and ligating the product in between the AgeI and NotI restriction sites of the α-actinin-tagRFP construct.

**Immunofluorescence.** Cells were seeded on glass coverslips (EMS) and left to adhere for 3 h. Cells were fixed in 1% (w/v) paraformaldehyde in RPMI medium for 30 min at 37 °C. 1% paraformaldehyde was always freshly prepared by adding 1% (w/v) paraformaldehyde to RPMI medium, heating it for 3 h at 65 °C and let it cool down to 37 °C. Next, cells were permeabilized in 0.1% (v/v) Triton X-100 in PBS for 5 min and samples were blocked with 2% (w/v) BSA in PBS with 20 mM glycine. The cells were incubated with primary Ab for 1 h, washed three times with PBS and incubated with secondary antibodies and phalloidin for 45 min. Samples were washed with phosphate buffer or MilliQ before embedding in Mowiol (Sigma-Aldrich).

**Structured illumination microscopy.** Structured illumination imaging was performed using a Zeiss Elyra PS1 system. 3D-SIM data was acquired using a ×63 1.4 NA oil objective. 488, 561, 642 nm 100 mW diode lasers were used to excite the fluorophores together with a BP 495–575 + LP 750, BP 570–650 + LP 750 or LP 655 filter, respectively. For 3D-SIM imaging the recommended grating was present in the light path. The grating was modulated in five phases and five rotations, and multiple z-slices with an interval of 110 nm were recorded on an Andor iXon DU 885, 1002 × 1004 EMCCD camera. Raw images were reconstructed using the Zeiss Zen 2012 software.

**Airyscan imaging.** Airyscan imaging was performed on a Zeiss LSM 880 system. 3D-Airyscan data was acquired using a ×63 1.4 NA oil objective. Laser characterics were 488 nm/25 mW, 561 nm/20 mW, or 633 nm/5 mW. Emission light was collected using a BP 420–480/BP 495–550, BP570–620/LP645 + SP615 and BP570–620/LP645 + LP660 for Alexa488, Alexa555/568, and Alexa647/633, respectively. Raw images were reconstructed using the Zeiss Zen 2.1 Sp1 software.

**STORM super-resolution microscopy.** Super-resolution microscopy was performed with a Leica SR GSD microscope (Leica Microsystems, Wetzlar, Germany) mounted on a Sumo Stage (#11888963) for drift-free imaging. Collection of images was done with an EMCCD Andor iXon camera (Andor Technology, Belfast, UK) and a ×160 oil immersion objective (NA 1.47). For the three-dimensional images an astigmatic lens has been used. To image, the samples have been immersed in the multi-color super-resolution imaging buffer OxEA[67]. Laser characteristics were 405 nm/30 mW, 488 nm/300 mW, and 647 nm/500 mW, with the 405 nm laser for back pumping. Ultra clean coverslips (cleaned and washed with base and acid overnight) were used for imaging. The number of recorded frames was variable between 10,000 and 50,000, with a frame rate of 100 Hz. The data sets were analyzed with the Thunder Storm analysis module[68], and images were reconstructed with a detection threshold of 70 photons, subpixel localization of molecules and uncertainty correction, with a pixel size of 10 nm.

**Electron microscopy.** For electron microscopy analysis of podosomes, DCs grown on PDMS 1:20 and 1:78 were washed with PBS, fixed in 1% GA in 0.1 M cacodylate (pH 7.4) buffer for 1 h at RT, washed and postfixed for 1 h at RT in 1% osmium tetroxide and 1% potassium ferrocyanide in 0.1 M cacodylate buffer. Cells were stained en bloc with 2% uranylacetate for 1 h at RT, washed with MQ, dehydrated in an ascending series of aqueous ethanol solutions and subsequently transferred via a mixture of ethanol and Durcupan to pure Durcupan (Sigma) as embedding medium according to standard procedures. Ultrathin grey sections (60–80 nm) were cut, contrasted with aqueous 2% uranyl acetate, rinsed and counterstained with lead citrate, air dried and examined in a JEOL JEM1400 electron microscope (JEOL) operating at 80 kV.

**DC transfection.** Transient transfections were carried out with the Neon Transfection System (Life Technologies). Cells were washed with PBS and resuspended in 115 µl resuspension buffer per 1 × 10$^6$ cells. Subsequently, cells were mixed with 7.5 µg DNA per 10$^6$ cells per transfection and electroporated. Directly after, cells were transferred to WillCo-dishes (WillCo Wells B.V.) with pre-warmed medium without antibiotics, serum, or phenol red. After 3 h, the medium was replaced by a medium supplemented with 10% (v/v) FCS and antibiotics. Before live-cell imaging, cells were washed with PBS and imaging was performed in HBSS supplemented with $Ca^{2+}$, $Mg^{2+}$, 5% (v/v) FCS and 25 mM HEPES. All live cell imaging was performed at 37 °C.

**Live cell imaging.** Live cell imaging for Supplementary Fig. 2 and Supplementary Movies 1 and 6 was performed on a Leica DMI6000 epifluorescence microscope equipped with an HC PL APO ×63/1.40–0.60 oil objective and a metal halide lamp. BFP was excited through a 360/40 nm band pass filter and emission was detected through a 475/40 nm band pass filter. GFP was excited through a 470/40 nm band pass filter and emission was detected through a 525/50 nm band pass filter. tagRFP was excited through a 546/12 nm band pass filter and emission was detected through a 605/75 nm band pass filter. iRFP was excited through a 620/60 nm band pass filter and emission was detected through a 700/75 nm band pass filter. Before live-cell imaging, cells were washed with PBS and imaging was performed in HBSS supplemented with $Ca^{2+}$, $Mg^{2+}$, 5% (v/v) FCS, and 25 mM HEPES. All live cell imaging was performed at 37 °C.

Airyscan live cell imaging for Figs. 4h–i, 5e–i, and Supplementary Movies 2–5 was performed on a Zeiss LSM 880, equipped with a PlanApochromatic ×63/1.4 NA oil immersion objective. The samples were excited with 488 nm argon (GFP) and 561 nm HeNe (mCherry) laser lines. Fluorescence emission was collected through a BP420–480 + BP495–550 (GFP), and a BP570–620 + LP645 (mCherry) filter. Time series were acquired with 15-s time interval. Emission signals for both channels were collected on the 32-channel GaAsP Airy detector. Before live-cell

imaging, cells were washed with PBS and imaging was performed in HBSS supplemented with Ca$^{2+}$, Mg$^{2+}$, 5% (v/v) FCS, and 25 mM HEPES. All live cell imaging was performed at 37 °C.

**PDMS substrate preparation.** Stiff (protrusion resistant) and soft (protrusion permissive) substrates were prepared using PDMS (Sylgard 184, Dow Corning). First, PDMS base and curing agent were thoroughly mixed in a 1:20 (stiff, ~800 kPa) and 1:78 (soft, ~1 kPa) ratio. These PDMS mixtures were subsequently used to coat WillCo-dishes (WillCo Wells B.V.) by spin coating 150 μl silicone mixture at 3100 rpm for 2 min. This resulted in thin (10–20 μm), high-resolution microscopy-compatible layers of PDMS. It is important to note that, while the optical lateral resolution on PDMS was similar compared to glass, the optical axial resolution decreased by ~1.5–1.7 fold (Supplementary Fig. 17).

**Fluorescence profile and height analysis.** To quantify the localization of each of these proteins with respect to the podosome core we used a semi-automatic self-developed ImageJ macro that (1) recognizes the podosome core centers based on the actin image, (2) draws a vertical line of ~3 μm through the center of the core that rotates around its center and collects a profile for every line, and (3) produces an average radial intensity profile as a function of distance from the podosome core center. Profiles are normalized to the minimum and maximum for visualization and comparison (see also Supplementary Fig. 1 for schematic overview). Of note, for some of the panels we only present one half of the intensity profile, since the radial intensity profiles are symmetric by definition. To quantify the features of the fluorescence intensity profile, the FWHM (WASP, Arp3, actin, α-actinin, γ-actin, and vinculin) and diameter (α-actinin, γ-actin, and vinculin) was determined using the Gaussian Fitting analysis tool of OriginPro 8. Only fits with an $R^2$ of >0.95 were accepted for analysis. Intensity features such as the dip in the α-actinin, γ-actin, and vinculin profile were not considered objective descriptives of the normalization procedure.

For the height analysis of some podosome components (actin, myosin IIA, vinculin). A similar approach was taken as described above but instead of generating a profile, an orthogonal view was collected for every line. For the height analysis, a line was drawn through the orthogonal views at a distance of 450 nm (vinculin), 800 nm (myosin IIA), 0 nm (core actin), and 800 nm (network actin) from the podosome core center. Fluorescence peak height for myosin IIA was calculated per cluster, since individual podosomes did not have sufficient myosin IIA signal to produce a reliable peak fit with OriginPro 8.

**Gelatin degradation assay.** Substrates were incubated for 20 min with 50 μg/ml poly-L-lysine (P2636, Sigma), washed three times in PBS and subsequently cross-linked with 0.25% (w/v) glutaraldehyde (Electron Microscopy Sciences) for 15 min. Next, substrates were washed three times with PBS and incubated with 25 μg/ml rhodamine-labeled gelatin for 30 min. Substrates were again washed three times with PBS prior to cell seeding. Cells were incubated for 16 h with the cells before fixation and staining with Alexa488-conjugated phalloidin. Gelatin degradation was assessed by calculating the average intensity of rhodamine fluorescence underneath the cells normalized to non-degraded areas.

**Sliding time window STICS analysis.** Sliding time window STICS analysis and subsequent PVC analysis was performed as described before[18]. We performed STICS[69] with a short time window iterated in single frame shifts on Airyscan time series of Lifeact-GFP and vinculin-mCherry, acquired with a 15 s time lag between frames. First, a Fourier immobile filter was applied in time to each pixel stack in the entire image series to remove the lowest frequency components[69]. Subsequently, each image was divided into 16 × 16 pixels ROIs (0.64 × 0.64 μm) and adjacent ROIs were shifted four pixels in the horizontal and vertical directions to map the entire field of view with oversampling in space. Time series were divided into overlapping 10 frame-sized TOIs (2.5 min) and adjacent TOIs were shifted one frame for each STICS analysis to cover the entire image series with oversampling in time. Space–time correlation functions were calculated for each ROI/TOI and fit for time lags up to τ = 10 to measure vectors (magnitude and direction) of the flow from the translation of the correlation peak as described earlier[70].

Detected noise vectors, which are due to random fits to spurious background peaks that pass multiple fitting threshold criteria, become more significant as we reduce the statistical sampling with short time windows. However, noise vectors exhibit little correlation with their neighbors in terms of direction and magnitude for systems where there are real flows. Due to the spatial and temporal oversampling (75% common overlap in space between adjacent ROIs and 90% common overlap in time between sequential TOIs) we expect neighboring vectors to correlate in magnitude and direction for real flows. Noise vectors that pass the fitting criteria were eliminated by setting a vector similarity criterion for adjacent vectors. All retained vectors were plotted on the corresponding frames of the immobile-filtered image series.

**Pair vector correlation.** To determine the spatial and temporal scales over which flow is correlated within a podosome cluster, we calculated the dot product between vectors separated in space and time. We calculated an average PVC

function for all pairs of vectors separated by the same spatiotemporal lags according to:

$$\text{PVC}(\delta r, \delta t) = \frac{1}{M_{\text{pairs}}(\delta r, \delta t)} \sum_i \sum_j v_i(r, t) \cdot v_j(r + \delta r, t + \delta t)$$

where $\delta r$ and $\delta t$ are the radial spatial and temporal lags, $M_{\text{pairs}}(\delta r, \delta t)$ denotes the number of vector pairs for each specified spatiotemporal lag and $v_i$ and $v_j$ are the vector pairs multiplied as dot products. When the angle between the two vectors lies between −90° and 90°, the dot product is positive. Conversely, when the angle between the two vectors is between −90° and −180° or 90° and 180°, the dot product is negative. When the vectors are uncorrelated the PVC will average to zero. For generating the graphs presented in Fig. 5h, i, the PVC result of five image series per condition were averaged.

**Statistics and reproducibility.** The type of statistical test, n values, and P values are all listed in the figure legends or in the figures. All statistical analyses (two-tailed Student's t-test) were performed using Graph Pad Prism or Microsoft Excel, and significance was determined using a 95% confidence interval. Non-linear contrast enhancement was applied to the actin images in Figs. 1d, e, 3a, d, e, 4a–c, 5a, b, and 6a, c, h to visualize the radiating actin filaments. Fluorescent profile generation and quantification of podosomes in those images was performed on the raw data before enhancement. Raw data are available upon request. All images were processed using Fiji[71] and figures were assembled in Microsoft Powerpoint. Box plots indicate median (middle line), 25th, 75th percentile (box) and 5th and 95th percentile (whiskers) as well as outliers (single points). All experiments were performed at least three times except for Figs. 1c, 2b–d, 6f, g, Supplementary Figs. 2, 4, and 7a, which were carried out two times. Supplementary Movie 1 is representative for three cells in two independent experiments, Supplementary Movies 2 and 3 are representative for five cells in three independent experiments, Supplementary Movie 6 is representative for five cells in two independent experiments. Statistical analysis was performed on the means of the experiments, except for Fig. 7g for which the means of the cells were compared, see also the Source Data file for the exact numbers and P values. It should be noted that all our data and plots were subjected to statistical analysis; in plots where statistical significance is not indicated, no statistically significant differences were found.

## Data availability

All primary data supporting the conclusions made are available from the authors upon request. The source data underlying Figs. 1a–c, f–h, 2b–d, 3b, c, 4d–h, 5c–f, 6a, b, 6d–g, 7b, d, g, k, and Supplementary Figs. 5b, c, 6b, 7a, b, 8c, d, 9, 10a are provided as a Source Data file.

## Code availability

All computer codes developed in Matlab and Fiji/ImageJ are made available from the authors upon request.

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

## Acknowledgements

We are indebted to the following people who kindly provided us the plasmids used in this manuscript: Johan de Rooij (vinculin-GFP); Michael Sixt (Lifeact-GFP); Stefan Linder (WASP-GFP); Dorothy Schafer (Arp3-GFP); Victor Small (vinculin-mCherry), Martin Schwartz (cortactin-BFP). The authors also thank the Microscopic Imaging Center of the Radboud Institute for Molecular Life Sciences and the Erasmus Optical Imaging Center for use of their microscopy facilities. M.A. is recipient of NWO-Veni 91615093 and a long-term fellowship (BUIT 2012-5347) from the Dutch Cancer Society. P.W.W. is recipient of a Natural Sciences and Engineering Research Council of Canada Discovery Grant. This work was financially supported by EU grant NANOVISTA (2882630) and by a Human Frontiers Science Program Grant (RGP0027/2012) to A.C.

## Author contributions

K.v.d.D. and B.J. performed experiments. J.A.S., A.M., and R.S. assisted with experiments. M.W. performed TEM imaging. L.N. performed STORM imaging. J.A.S. performed SIM imaging. M.B.M.M. and E.P. performed twSTICS and PVC analysis. M.A. provided mouse BMDCs. J.F., A.B.H., P.W.W., and K.J. provided analytical tools. K.v.d.D. carried out data analysis and prepared all the figures. K.v.d.D. and A.C. designed the experiments and wrote the manuscript with input from all authors. A.C. supervised the study.

## Competing interests

The authors declare no competing interests.
