## [Peer Review File · Nature Communications]

Reviewers' comments:

Reviewer #2 (Remarks to the Author):

In this study, the authors investigate internal organization of podosomes in dendritic cells using a variety of high-resolution light microscopy techniques. This revealed a structural feature that consists of the innercore, cCAM, and outer core, pCAM, that contains different subsets of actin isoforms and actin binding proteins. The cCAM/pCAM organization was shown to be stiffness-independent. On the other hand, stiffness was shown to affect actin filaments that radiate from the podosomes. On stiff substrate, podosome oscillation was observed, while this was damped on soft substrate. Similarly, on stiff substrate, matrix degradation was observed, but this was much reduced on soft substrate. The authors suggest that this represents a stiffness-induced switch in nanoscale architecture of podosomes, which control their mesoscale activity and degradative capacity.

Overall, while this study presented new information on intra-podosome organization and high-quality experimental results and quantitative analysis befitting the long-standing expertise of the authors, this reviewer has reservations on the following aspects: 1) The claim of the authors (i.e. the title) does not seem to be adequately supported by the data as presented; 2) novelty; 3) physiological relevance; 4) mechanistic insights, as described further below.

1) The author's claim regarding "stiffness-induced switch in nanoscale architecture of podosomes" does not seem to be a proper description of the observed phenomenon. First, the cCAM/pCAM organization of the podosome core was shown to be largely conserved regardless of stiffness (Fig. 3). Second, the most notable change is the radiating filament organization (fig. 4a&g). However, it is quite challenging to observe these changes, given the image quality, even though the authors themselves have presented better quality data of podosomes in their MBoC 2012 or Nature Comms 2016 papers. Third, the term "nanoscale" would suggest ultrastructural changes, but these changes seem to occur at a somewhat larger scale. Here, perhaps an EM-based technique such as in Luxenburg..Addadi, Plos One 2007, might be useful to corroborate what the authors observed here. Fourth, stiffness is a continuous property but it was presented as if it is a binary quantity (soft/stiff). To ascertain that this is a "switch" in response to stiffness, it would be important to have a graded variation in stiffness, and data points from these intermediate stiffness values. In other words, the authors only have two data points regarding the stiffness. This could very well be a gradual monotonic change instead of a "switch" as claimed by the authors—two data points are not enough to differentiate between the two possibilities.

2) Novelty: although the author presented new information on sub-podosome organization, a number of key findings i.e. change in the radiating filament organization seems to be incremental, especially given authors' previous works which already described the importance of Myosin IIA contractility, and substrate topography. Thus, podosomes have been known to be mechanosensitive and thus their dependence on stiffness is not surprising. Likewise, the quantitative analysis methods used here have already been described by the authors previously, while a detailed analysis of podosome nanoscale architecture was also recently presented by Bouissou et al, ACS Nano 2017 (the modelling aspect of this paper was cited, but whether the nanostructural findings there are similar or different to what the authors found here was not discussed).

3) Physiological relevance: Although the oscillatory behaviors of podosomes are interesting from the point of view of single cell biophysics, the observation that it is largely absent on soft substrate raises a question regarding their physiological relevance. The basement membrane or any other tissues (except for the bones perhaps) that the podosomes degrade in physiological condition would be much softer compared to glass. On a related note, what is the elastic modulus of the PDMS was not clearly indicated, and how are these related to physiological stiffness of various tissues?

4) Mechanistic insights: The authors's data on stiffness-dependent switch is presented in a largely descriptive manner. What are the necessary molecules, protein-protein interactions, or signalling pathways that mediate this so-called "switch"? What mediate cross-talk between gelatin degradation (presumably vesicular transport-dependent) and podosome dynamics? Are they co-regulated or independently-regulated? If these mechanistic aspects are further elaborated, this would strengthen this manuscript significantly.

Reviewer #3 (Remarks to the Author):

The manuscript by van den Dries et al. 'Stiffness-induced switch in nanoscale architecture controls podosome mesoscale connectivity and degradative capacity' deals with the question of how podosome nanoscale architecture support its functions including mechano-sensing and degradative capacity. Using super-resolution microscopy techniques (SIM, Airyscan imaging, dSTORM) the authors studied the nanoscale organization of actin isoforms (beta, gamma), actin binding and regulatory proteins (alpha-actinin, Arp2/3, WASP, cortactin), and integrin adhesion associated proteins (vinculin, zyxin, talin) in podosomes of primary human dendritic cells. They demonstrate that podosomes display a multi-modular architecture composed of a central Core Actin Module (cCAM) and a peripheral Core Actin Module (pCAM). Arp3 and WASP localize to the cCAM while alpha-actinin localizes to the pCAM. Beta-actin and gamma-actin isoforms localize to cCAM and pCAM, respectively. After description of the nanoscale architecture of podosomes, the authors studied podosome mechano-sensing using substrates (PDMS) of different rigidities. Their results suggest that cCAM and pCAM are not affected by changes in substrate stiffness. However, substrate stiffness controls radiating actin filament organization, podosome mesoscale connectivity and degradative capacity. The main conclusion is that podosomes switch from a long range connected state associated with degradative behavior to a locally connected state associated with protrusion.

The manuscript contains interesting findings on podosome nanoscale organization and mechano-sensing, which the authors try to correlate. However, the links between the multimodular organization of podosomes and mechano-sensing of podosomes are not demonstrated by the results. Indeed the stiffness of the substrate does not seem to affect the nanoscale organization of the podosome (Fig. 3). Exploration of myosin II involvement in podosome mechano-sensing is missing, and could bring important insights into the molecular mechanisms involved. At the exception of the main point raised above, experimentally the manuscript is solid and contains interesting finding that will stimulate further experiments in the field.

Major concerns that should be addressed:

1. The manuscript appears to contain two parts that are not experimentally linked: (1) description of podosome cores which consist of a central branched actin module (cCAM) and a peripheral linear actin module (pCAM), each binding specific actin interactors and actin isoforms; (2) effects of substrate rigidity on podosomes: softer substrates leading to decreased density of radiating actin filaments, reorganization of tension sensitive adaptor proteins (vinculin, zyxin), reduced mesoscale connectivity and degradative capacity. It will be very interesting to link those two parts.

One possibility could be to decrease even more PDMS rigidity to test if this will finally affect podosome nanoscale architecture. Indeed, in the described experiments cCAM and pCAM seems unaffected by PDMS rigidity. However, on softer substrates the protrusion of the actin core should be increased and the counteracting forces of the substrate should be decreased. This would probably affect the nanoscale architecture of the podosome, in light of the schematic described in Supplementary figure 12. Could the authors try softer substrates? This could increase the length of the protrusive actin core to a detectable level; this could also lead to a decreased diameter of the

dome in the optical axis. Those results will definitively link the podosome nanoscale architecture to mechano-sensing.

Related to the same comments: Could the authors explain why the lower part of the actin core, which should correspond to actin protrusion, is not visible in Fig. 2f and Fig. 3c? The authors measured the FWHM and diameters (XY) of actin, alpha-actinin and WASP on substrates of different rigidities. They should also measure the heights of cCAM and pCAM (diameters in the optical axis (Z axis) for alpha-actinin, heights for WASP and actin) to test if lower stiffness affects the height of the actin dome.

2. Lowering substrate rigidity seems to recapitulate some effects found when myosin II activity is inhibited (see from the Cambi's group: Actomyosin-dependent dynamic spatial patterns of cytoskeletal components drive mesoscale podosome organization. Nature Communications 2016): (1) decreased density of radiating F-actin; (2) decreased actin velocity; (3) decreased density of actin movements that could correspond to lower connectivity between podosomes. Furthermore, inhibiting myosin II contractility is associated with decreased degradative function (cited in the discussion: Alexander et al. Extracellular matrix rigidity promotes invadopodia activity. Current biology 2008), which fits with the decreased degradation observed on softer substrates.

Thus, could it be that the main effect of lowering substrate rigidity is in fact funneling down to decreased contractility of the cell? It could be very interesting to test how decreasing (e.g. blebbistatin or other inhibitors...) and increasing (e.g. Calyculin A, RhoA mutants...) myosin II activity affect podosome response to changes in substrate stiffness. For example, treating the cells with calyculin A could prevent the effects observed on soft substrates, demonstrating that MII activity is tune down on soft substrates. This could, for example, support a mechanism where on soft substrate myosin II need to be turned down to disassemble radiating F-actin enabling the transition from an explorative, degradative behavior, to a focally protrusive, non-degradative state.

In any case, I think that it is critical to understand the role played by myosin II during rigidity-sensing of podosomes.

3. How the authors could measure the length of F-actin radiating filaments. In the presented figures, F-actin looks more like bundles that are densely packed in between podosomes. Could the authors describe precisely how they performed their measurements and analysis?

Minor concerns:

1. WASP is not associated with the branched actin network per se, but it is rather activating the Arp2/3 complex. Thus WASP should be more associated with the membrane at the base the podosome. It could be interesting to perform SIM on Arp3 and WASP to test this.

2. Could it be that alpha-actinin is more associated with the adhesive part of the podosome. The authors say p. 6 'alpha-actinin localization is markedly different from the localization of the ring component vinculin' but the diameter of alpha-actinin, at the base of the dome, does not seem different that the one of vinculin (Fig. 1e). Thus, I am not totally convinced that alpha-actinin is more associated with the core actin than with the adhesive ring. Actually, in the schematic of Fig. 2g, the authors associate pCAM with the adhesive ring. Could the authors comment on this.

Reviewer #2 (Remarks to the Author):

In this study, the authors investigate internal organization of podosomes in dendritic cells using a variety of high-resolution light microscopy techniques. This revealed a structural feature that consists of the inner core, cCAM, and outer core, pCAM, that contains different subsets of actin isoforms and actin binding proteins. The cCAM/pCAM organization was shown to be stiffness-independent. On the other hand, stiffness was shown to affect actin filaments that radiate from the podosomes. On stiff substrate, podosome oscillation was observed, while this was damped on soft substrate. Similarly, on stiff substrate, matrix degradation was observed, but this was much reduced on soft substrate. The authors suggest that this represent a stiffness-induced switch in nanoscale architecture of podosomes, which control their mesoscale activity and degradative capacity.

Overall, while this study presented new information on intra-podosome organization and high-quality experimental results and quantitative analysis befitting the long-standing expertise of the authors, this reviewer has reservations on the following aspects: 1) The claim of the authors (i.e. the title) does not seem to be adequately supported by the data as presented; 2) novelty; 3) physiological relevance; 4) mechanistic insights, as described further below.

General response: We appreciate the thorough evaluation of the reviewer and we took his/her reservations very seriously to improve our study. We have considerably revised our manuscript by performing additional experiments to enrich and strengthen the message. This resulted in 3 new Supplementary Figures and 2 new main Figures. More specifically, we increased the novelty by adding the discovery of different actin networks in between podosomes and addressed the physiological relevance by better showing the behavior of podosomes as a function substrate stiffness. Altogether, this resulted in a thoroughly revised manuscript that in our view now much better provides mechanistic insight into the podosome structure-function relationship. We hope the reviewer will concur that the data presented in the revised manuscript now do support our claims. Below, for clarity, we address each point by providing an answer to the specific remark (**response to the reviewer**) and/or an explanatory text on the related revisions (**revised manuscript**). In the revised manuscript, we have highlighted the new/revised sections in yellow.

NB. To prevent confusion it is important to note that, in the revised manuscript, we renamed the submodules of the core to better reflect their function.

The pCAM is now *pPM* (*peripheral protrusion module*)

The cCAM is now *cPM* (*central protrusion module*)

1) The author's claim regarding "stiffness-induced switch in nanoscale architecture of podosomes" does not seem to be a proper description of the observed phenomenon.

Response to reviewer: The reviewer raises valid criticisms related to the main claim in the title of the original manuscript. Our original manuscript appeared to consist of two parts that were insufficiently linked, something that was also pointed out by Reviewer #3. In the first part, we presented novel findings on the nanoscale architecture of podosome core subregions and in the second part, we showed that substrate stiffness does not affect the organization of these subregions but rather that of the radiating filaments. For the original manuscript, we tried to catch these two parts in one title. The "switch in the nanoscale architecture of podosomes" in the original title did not refer to the core subregions but to the changes in the actin filaments. It is, however, clear from the reviewer's comment that our original title was far from optimal. The revised manuscript now addresses more clearly the complexity of the

podosome structure in relation to its pushing behavior on soft and stiff substrates. To better reflect the new content and sharpen the key message of our current study, we have designed a new title for the revised manuscript: “*Modular actin nano-architecture enables podosome protrusion and mechanosensing*”.

First, the cCAM/pCAM organization of the podosome core was shown to be largely conserved regardless of stiffness (Fig. 3). Second, the most notable change is the radiating filament organization (fig. 4a&g). However, it is quite challenging to observe these changes, given the image quality, even though the authors themselves have presented better quality data of podosomes in their MBoC 2012 or Nature Comms 2016 papers.

Response to reviewer: Our initial data indeed indicated that the subregions in the podosome core do not respond to changes in substrate stiffness, and that the organization of the radiating filaments is changed. Regarding the image quality of the current data with respect to our previous publications, we would like to point out that the visibility of the actin filaments in podosome images is notoriously difficult since the filaments in between podosomes are much dimmer compared to the intense actin signal of the podosome cores. The extent of this visibility problem strongly depends on the actin probe (phalloidin, lifeact) as well as on the imaging technique, especially nowadays with the many reconstruction algorithms for super-resolution microscopy. As an example, for the dSTORM images in our 2012 MBoC publication, we solved this particular issue with a non-linear lookup table. Yet, in the original manuscript, based on the quality of the data from the Airyscan, we did not find this necessary and opted for a linear lookup table in combination with a minor contrast enhancement. Yet, it is clear from the reviewer’s comment that the changes in actin filament organization were difficult to observe in the original images.

Revised manuscript:

This criticism prompted us to perform a more thorough nanoscale imaging of the various actin filaments detected in the podosome area which resulted in a substantial amount of new data that **1)** reveal novel features of the actin architecture in between podosomes and **2)** better describe the podosome response to stiffness. We believe that these new datasets also add to the novelty aspect of our manuscript, as we explain below. Furthermore, we have improved the image quality as explained below in more detail.

*First of all, we now present 3D super-resolution data demonstrating that the actin network in between podosomes actually consists of two networks at different heights: one is more localized at the ventral side of the podosome cluster (close to the plasma membrane) and is associated to vinculin, and the other one is more on the dorsal side of the podosome cluster (500 nm above the plasma membrane) and crosslinked by myosin IIA. These data are now presented in the **new Figure 3** and **new Supplementary Figure 5** and described on **page 9-12** of the revised manuscript.*

*Second, we further substantiate our initial results that the podosome core subregions do not respond to changes in substrate stiffness by showing that the ratio between beta and gamma actin isoforms is similar on stiff and soft substrates as measured by both immunofluorescence as well as western blot analysis. These results are now presented in the **revised Supplementary Figure 9** and described on **page 14** of the revised manuscript.*

*Third, we show that the dorsal filaments crosslinked by myosin IIA do not appear to respond to substrate stiffness. In the **new Figure 5** and **Supplementary Figure 10**, we now show that myosin IIA*

localization and activation does not change in response to substrate stiffness, suggesting that the dorsal network is not *the* mechanoresponsive element. These results are described on **page 14-15** of the revised manuscript.

Finally, to overcome the image quality issue mentioned above, we have now applied a non-linear contrast enhancement (gamma correction) to all of our actin images (as now also mentioned in the figure legends). In this way, the intensity of the dim filaments is enhanced without saturating the signal of the podosome cores. We hope that the reviewer will now appreciate the differences in the organization of the ventral filaments that is represented in the **revised Figures 6a, c and h** (Original Figure 4).

Overall, we hope the reviewer will agree that the revised manuscript presents a more detailed analysis and clear representation of the actin filament organization in between podosomes. Importantly, these revisions have also substantially contributed to the coherency of the story presented in the manuscript. In the first part of the manuscript (**Figures 1-3**) we now present data that identifies subregions within the core and reveals the existence of two different actin networks (summarized in the model in the **new Figure 3e**). In the second part (**Figures 4-7**) we show that specifically the ventral filaments, and not the dorsal filaments or the core subregions, respond to substrate stiffness which correlates with different dynamic behavior as well as degradative capacity of podosomes (summarized in the model in the **revised Supplementary Figure 16**).

Third, the term “nanoscale” would suggest ultrastructural changes, but these changes seem to occur at a somewhat larger scale. Here, perhaps an EM-based technique such as in Luxenburg..Addadi, Plos One 2007, might be useful to corroborate what the authors observed here.

Response to reviewer: In response to substrate stiffness, we observed changes in the length of the actin filaments (difference of ~170 nm) and localization of vinculin (difference in width of ~100 nm, difference in diameter of ~100 nm). These changes are well in the submicron range, which in our view justifies the use of the term ‘nanoscale’. Regarding the use of an EM-based technique, we have extensively tried to image the PDMS substrates by Scanning EM (SEM) to visualize the filaments in greater detail. Yet, this appeared technically challenging especially with the soft PDMS, where we noticed some non-crosslinked PDMS covering the sample probably being released during the dehydration steps. Another reason to not use the SEM to confirm the fluorescent images is the fact that by unroofing the cells, we entirely lose the dorsal filament network, based on myosin IIA staining. These observations were obtained as part of a study using Correlative Light and Electron Microscopy to determine localization of the adaptor proteins vinculin and zyxin with respect to the actin architecture recently published by our group (*see Joosten et al Front Immunol 2018*).

Fourth, stiffness is a continuous property but it was presented as if it is a binary quantity (soft/stiff). To ascertain that this is a “switch” in response to stiffness, it would be important to have a graded variation in stiffness, and data points from these intermediate stiffness values. In other words, the authors only have two data points regarding the stiffness. This could very well be a gradual monotonic change instead of a “switch” as claim by the authors—two data points are not enough to differentiate between the two possibilities.

Response to reviewer: We realize that by using the term “switch”, we may have given the wrong impression that we consider stiffness as a binary quantity. We can reassure the reviewer that this is not the case. In fact, we have several datasets (i.e. on the length of the actin filaments, the relocalization

vinculin as well as the gelatin degradation) obtained from cells adherent onto PDMS substrates of increasing intermediate stiffness (i.e.: ~1kPa, ~12kPa, ~30kPa and ~800kPa) and indeed showing a gradual effect of stiffness on podosome remodeling and degradative activity (see **Figure A** below). We, however, left these data out of the original manuscript as we only wanted to highlight the structural changes that occur in response to *very stiff*, non-compliant substrate and a *very soft*, compliant substrate that is deformed by podosome protrusive forces. As shown in **Supplementary Figure 8**, this is the case for the two different formulations of PDMS that we used. We show these intermediate stiffness data here to the reviewer as a confidential Figure since we believe they are not necessary in the current manuscript. We are willing to include them as supplementary information to the revised manuscript, should the reviewer or the editor wish so. In any case, we have now removed the misleading term “switch” from both the title as well as the main text of the manuscript to prevent any confusion.

Figure A for Reviewer

2) *Novelty: although the author presented new information on sub-podosome organization, a number of key findings i.e. change in the radiating filament organization seems to be incremental, especially given authors' previous works which already described the importance of Myosin IIA contractility, and substrate topography. Thus, podosomes have been known to be mechanosensitive and thus their dependence on stiffness is not surprising. Likewise, the quantitative analysis methods used here have already been described by the authors previously, while a detailed analysis of podosome nanoscale architecture was also recently presented by Bouissou et al, ACS Nano 2017 (the*

modelling aspect of this paper was cited, but whether the nanostructural findings there are similar or different to what the authors found here was not discussed).

Response to reviewer: We have carefully considered this criticism on limited novelty of our original findings in relation to our previously published work. Although the original manuscript extended and built upon those previous results, it also put forward new concepts in podosome biology, which included: **1)** the differential localization in actin binding proteins (alpha-actinin, Arp2/3, WASP) in the podosome core, **2)** the differential localization of actin isoforms in the podosome core, **3)** the reorganization of actin filaments as a function of substrate stiffness, **4)** the differential dynamic behavior as a function of substrate stiffness and **5)** the decreased matrix degradation of podosomes as a function of substrate stiffness. Of course, podosomes have been shown to be mechanosensitive, but the structural organization driving this property was still unclear. Our study was (and still is) meant to fill this gap. In any case, the reviewer's remark triggered us to expand and strengthen our studies as we further discuss below.

In this remark, the reviewer also stated he/she missed a thorough comparison of our results with the work by *Bouissou et al in ACS Nano 2017*. That publication nicely describes the 3D/axial organization of talin and vinculin around podosomes. Our focus, however, is entirely on the modular architecture of actin at individual podosomes and in the cluster. Therefore, we limited our discussion to the modelling aspect of the *ACS Nano* article since we feel that element related most to our experimental data.

Revised manuscript: As already mentioned above, we have now added a substantial amount of data that provides additional novelty to the original manuscript. First and foremost, we now present 3D super-resolution data that indicate the presence of two networks in between podosomes, a ventral network that is associated with vinculin and a dorsal network that is associated with myosin IIA (**new Figure 3** and the **new Supplementary Figure 5**). Although we were aware that vinculin and myosin IIA localize to the filaments in between podosomes, we had never been able to uncover their respective localization. This knowledge gap now seems to be filled by our observations that they associate to pools of actin localized at different heights in between podosomes. These findings not only foster an entirely new concept for podosome architecture, but they also nicely put some of our previously published results into perspective (e.g. the fact that myosin IIA inhibition does not affect vinculin localization, as reported in *Nat Comm 2013 and 2016*, is explained by the fact that vinculin and myosin IIA are enriched at different heights within the radiating actin filaments). Furthermore, we now show that myosin IIA localization and activation are not controlled by substrate stiffness (**new Figure 5** and **new Supplementary Figure 10**), while the ventral actin filaments and associated vinculin are (**revised Figure 6**). The fact that myosin IIA seems dispensable for podosome stiffness sensing is not only interesting in the context of podosomes but also in relation to other actin-based structures such as focal adhesions, where myosin IIA has been shown to control their mechanoresponsiveness.

Altogether, we believe that our additional findings, in combination with the data presented in the original manuscript, present many novel aspects of podosome biology and actin spatiotemporal organization. Our findings put forward a new paradigm for force generation and transmission at these cellular protrusions. Based on our discovery of this modular actin architecture of podosomes, we propose

that the classical core-ring model inadequately explains podosome force generation and, for the first time, present a fully integrated structure-function model for how protrusion and mechanosensing may be regulated by podosomes. This new model and its implications for force transmission are presented in the **revised Supplementary Figure 16** and discussed on **page 24-25** of the revised manuscript.

3) Physiological relevance: Although the oscillatory behaviors of podosomes are interesting from the point of view of single cell biophysics, the observation that it is largely absent on soft substrate raise question regarding their physiological relevance. The basement membrane or any other tissues (except for the bones perhaps) that the podosomes degrade in physiological condition would be much softer compared to glass. On a related note, what is the elastic modulus of the PDMS was not clearly indicated, and how are these related to physiological stiffness of various tissues?

Response to reviewer: Apparently, our text has been misleading as the reviewer had the impression that podosome oscillations are “largely absent” on soft substrates, which possibly fostered her/his skepticism on the physiological relevance. In reality, we showed that actin and vinculin undergo concerted oscillations both on soft and stiff substrates (**original Figure 4h, i; revised Supplementary Figure 7**). We did, however, show that the dynamic spatial patterns and mesoscale connectivity are different on soft and stiff substrates. We interpreted our results such that podosomes on stiff surfaces have to scan relatively large areas for ‘weak’ spots while actively degrading the surface. Once the surface softens, podosomes turn to a focally protrusive, non-degradative state that enables protrusion. We believe our findings go beyond single cell biophysics as the physiological relevance of our study resides in providing a better understanding of how the actin-based protrusions enables cells to sense and breach tissue barriers.

Regarding the elastic modulus of the PDMS, this information was inadvertently left out during the writing process of the original manuscript. We apologize for this omission. We used two curing to base ratios of PDMS: 1:20 which corresponds to ~800kPa (stiff) and 1:78 which corresponds to ~1kPa (soft). This information is now also added to the results section on **page 12** and to the materials and methods section on **page 33** of the revised manuscript.

4) Mechanistic insights: The authors’s data on stiffness-dependent switch is presented in a largely descriptive manner. What are the necessary molecules, protein-protein interactions, or signalling pathways that mediate this so-called “switch”? What mediate cross-talk between gelatin degradation (presumably vesicular transport-dependent) and podosome dynamics ? Are they co-regulated or independently-regulated? If these mechanistic aspects are further elaborated, this would strengthen this manuscript significantly.

Response to reviewer: Extensive microscopy studies often suffer from the misconception of only providing descriptive data, as remarked here by the reviewer. We disagree with this view, since dissecting the structural determinants that enable complex cytoskeletal structures to perform their function is also a way to provide mechanistic information. Although our original study already provided novel mechanistic information on how podosomes protrusive forces are generated, we do agree with the reviewer that the manuscript would have benefitted from a stronger mechanistic message. Below we discuss the new experiments we now included in the revised manuscript.

Revised manuscript: As mentioned above, the revised manuscript now reveals the modular actin nano-architecture that enables podosome protrusion and mechanosensing. Specifically, we show that:

- 1)** the podosome protrusive core contains a central branched actin module encased by a linear actin module, each harboring specific actin interactors and actin isoforms;
- 2)** from the core, two actin modules radiate: ventral filaments bound by vinculin and connected to the plasma membrane and dorsal interpodosomal filaments crosslinked by myosin IIA.
- 3)** on stiff substrates, the actin modules mediate long-range substrate exploration, associated with degradative behavior. On compliant substrates, the vinculin-bound ventral actin filaments shorten, resulting in short-range connectivity and a focally protrusive, non-degradative state.

To uncover additional mechanistic aspects with respect to the regulation of the stiffness response by podosomes, we thoroughly investigated both the localization and activity of myosin IIA at podosomes formed on soft and stiff substrates. Strikingly, we did not observe any changes in both the localization as well as the activity (as measured by pMLC in immunofluorescence) when podosomes were formed on soft or stiff substrates (**new Figure 5** and **new Supplementary Figure 10**). To further support the notion that myosin IIA does not play a major role in podosome stiffness sensing, we inhibited myosin IIA on both stiff and soft substrates and found that the changes in the relocalization of vinculin are not altered (**Supplementary Figure 12**). Our data therefore support a model where podosome stiffness sensing is independent from myosin IIA activity. As now discussed at **page 26**, we propose that changes in actin polymerization on soft and stiff substrates could drive podosome stiffness sensing.

We believe that the substantial amount of new experimental evidence in the revised manuscript now redefines podosome nanoscale architecture and reveals a paradigm for how actin modularity drives invadosome mechanosensing in cells that breach tissue boundaries. We hope the reviewer will agree that it is outside of the scope of this work to further unravel the signaling mechanisms that regulate podosome stiffness sensing.

Reviewer #3 (Remarks to the Author):

The manuscript by van den Dries et al. 'Stiffness-induced switch in nanoscale architecture controls podosome mesoscale connectivity and degradative capacity' deals with the question of how podosome nanoscale architecture support its functions including mechano-sensing and degradative capacity. Using super-resolution microscopy techniques (SIM, Airyscan imaging, dSTORM) the authors studied the nanoscale organization of actin isoforms (beta, gamma), actin binding and regulatory proteins (alpha-actinin, Arp2/3, WASP, cortactin), and integrin adhesion associated proteins (vinculin, zyxin, talin) in podosomes of primary human dendritic cells. They demonstrate that podosomes display a multi-modular architecture composed of a central Core Actin Module (cCAM) and a peripheral Core Actin Module (pCAM). Arp3 and WASP localize to the cCAM while alpha-actinin localizes to the pCAM. Beta-actin and gamma-actin isoforms localize to cCAM and pCAM, respectively. After description of the nanoscale architecture of podosomes, the authors studied podosome mechano-sensing using substrates (PDMS) of different rigidities. Their results suggest that cCAM and pCAM are not affected by changes in substrate stiffness. However, substrate stiffness controls radiating actin filament organization, podosome mesoscale connectivity and degradative capacity. The main conclusion is that podosomes switch from a long range connected state associated with degradative behavior to a locally connected state associated with protrusion.

The manuscript contains interesting findings on podosome nanoscale organization and mechano-sensing, which the authors try to correlate. However, the links between the multimodular organization of podosomes and mechano-sensing of podosomes are not demonstrated by the results. Indeed the stiffness of the substrate does not seem to affect the nanoscale organization of the podosome (Fig. 3). Exploration of myosin II involvement in podosome

mechano-sensing is missing, and could bring important insights into the molecular mechanisms involved. At the exception of the main point raised above, experimentally the manuscript is solid and contains interesting finding that will stimulate further experiments in the field.

General response to reviewer: We are grateful to the reviewer for her/his positive evaluation of our work and constructive suggestions. We agree that our initial manuscript had two shortcomings.

First, the manuscript appeared to consist of two parts that were insufficiently linked: a first part, in which we presented novel findings on the nanoscale architecture of podosome core subregions and a second part, where we showed that substrate stiffness specifically affects the radiating filaments. We now added additional experimental data and restructured the manuscript. In the first part (**Figures 1-3**), we now present data that identifies subregions within the core and reveals the existence of two different actin networks (summarized in the model in the **new Figure 3e**). In the second part (**Figures 4-7**) we show that specifically the ventral filaments, and not the dorsal filaments or the actin subregions, respond to stiffness which correlates with different dynamic behavior as well as degradative capacity of podosomes (summarized in the model in the **revised Supplementary Figure 16**). We hope the reviewer will agree that this has significantly improved the manuscript coherence.

Second, in the original manuscript, we had chosen to purely focus on actin and excluded myosin IIA. However, we agree with the reviewer that the exploration of myosin IIA involvement in podosome mechanosensing would have added important novel insights. We therefore devoted the past year to thoroughly investigate this aspect, which now significantly contributes to the main message of the manuscript: see **new Figure 3** and **new Figure 5**.

Below, for clarity, we address each point by providing an answer to the specific remark (**response to the reviewer**) and/or an explanatory text on the related revisions (**revised manuscript**). In the revised manuscript, we have highlighted the new/revised sections in yellow.

NB. To prevent confusion it is important to note that, in the revised manuscript, we renamed the submodules of the core to better reflect their function.

The pCAM is now *pPM* (*peripheral protrusion module*)

The cCAM is now *cPM* (*central protrusion module*)

Major concerns that should be addressed:

1. The manuscript appears to contain two parts that are not experimentally linked: (1) description of podosome cores which consist of a central branched actin module (cCAM) and a peripheral linear actin module (pCAM), each binding specific actin interactors and actin isoforms; (2) effects of substrate rigidity on podosomes: softer substrates leading to decreased density of radiating actin filaments, reorganization of tension sensitive adaptor proteins (vinculin, zyxin), reduced mesoscale connectivity and degradative capacity. It will be very interesting to link those two parts.

One possibility could be to decrease even more PDMS rigidity to test if this will finally affect podosome nanoscale architecture. Indeed, in the described experiments cCAM and pCAM seems unaffected by PDMS rigidity. However, on softer substrates the protrusion of the actin core should be increased and the counteracting forces of the substrate should be decreased. This would probably affect the nanoscale architecture of the podosome, in light of the schematic described in Supplementary figure 12. Could the authors try softer substrates? This could increase the length of the protrusive actin core to a detectable level; this could also lead to a decreased diameter of the dome in the optical axis. Those results will definitively link the podosome nanoscale architecture to mechano-sensing.

Response to reviewer: We agree that our original manuscript consisted of two parts that were insufficiently linked. A similar concern was raised by reviewer #2, who felt that the title did not properly reflect the two parts of the manuscript. To better link the two parts of the manuscript, we have now added a substantial amount of data that **1)** reveals novel features of the architecture of the filaments in between podosomes and **2)** better describes the stiffness response of podosomes.

Revised manuscript: First of all, we now present 3D super-resolution data that indicate that the network in between podosomes actually consists of two networks; one of which is on the ventral side of podosome clusters and associated to vinculin, and the other which is on the dorsal side and crosslinked by myosin IIA. These data are now presented in the **new Figure 3** and **new Supplementary Figure 5** and described on **page 9-12** of the revised manuscript.

Second, we further substantiate our initial results that the podosome core subregions do not respond to changes in substrate stiffness by showing that the ratio between beta and gamma actin is similar on stiff and soft substrates as measured by both immunofluorescence as well as western blot analysis. These results are now presented in the **revised Supplementary Figure 9** and described on **page 14** of the revised manuscript.

Third, we show that the dorsal filaments do not appear to respond to substrate stiffness. In the **new Figure 5** and **Supplementary Figure 10**, we show that myosin IIA localization does not change in response to substrate stiffness, suggesting that the dorsal network is not mechanoresponsive. These results are described on **page 14-15** of the revised manuscript.

The reviewer suggests the use of even softer surfaces to study podosomes. Unfortunately, the curing to base ratio (cure:base = 1:78) that we used in the manuscript results in the lowest stiffness (~1 kPa) that you can create with PDMS. At even lower ratios, the PDMS will not cure, remains fluid and cannot be used for cell adhesion studies. In any case, based on the publication by *Labernadie et al in Nat Comms 2014*, which described podosome protrusion depth of ~20 nm in soft materials by AFM, we do not expect much more indentation in even softer materials. Podosomes most likely do not have the ability to generate deeper protrusions into any type of substrate apart from degradable biological tissues.

Related to the same comments: Could the authors explain why the lower part of the actin core, which should correspond to actin protrusion, is not visible in Fig. 2f and Fig. 3c? The authors measured the FWHM and diameters (XY) of actin, alpha-actinin and WASP on substrates of different rigidities. They should also measure the heights of cCAM and pCAM (diameters in the optical axis (Z axis) for alpha-actinin, heights for WASP and actin) to test if lower stiffness affects the height of the actin dome.

Response to reviewer: The reviewer suggests to measure the FWHM in z for several core components. We deliberately did not measure the FWHM in z for actin, WASP and arp2/3, since the z resolution is worsened on PDMS because of an elongated PSF (see also **Supplementary Figure 17**). This worsened z resolution (800-1000 nm), in combination with the very small indentations that we observed by TEM (~80 nm), have refrained us from making claims about the FWHM in z.

2. Lowering substrate rigidity seems to recapitulate some effects found when myosin II activity is inhibited (see from the Cambi's group: Actomyosin-dependent dynamic spatial patterns of cytoskeletal components drive mesoscale podosome organization. Nature Communications 2016): (1) decreased density of radiating F-actin; (2) decreased actin velocity; (3) decreased density of actin movements that could correspond to lower connectivity between

podosomes. Furthermore, inhibiting myosin II contractility is associated with decreased degradative function (cited in the discussion: Alexander et al. Extracellular matrix rigidity promotes invadopodia activity. Current biology 2008), which fits with the decreased degradation observed on softer substrates.

Thus, could it be that the main effect of lowering substrate rigidity is in fact funneling down to decreased contractility of the cell? It could be very interesting to test how decreasing (e.g. blebbistatin or other inhibitors...) and increasing (e.g. Calyculin A, RhoA mutants...) myosin II activity affect podosome response to changes in substrate stiffness. For example, treating the cells with calyculin A could prevent the effects observed on soft substrates, demonstrating that MII activity is tune down on soft substrates. This could, for example, support a mechanism where on soft substrate myosin II need to be turned down to disassemble radiating F-actin enabling the transition from an explorative, degradative behavior, to a focally protrusive, non-degradative state.

In any case, I think that it is critical to understand the role played by myosin II during rigidity-sensing of podosomes.

Response to reviewer: We agree with the reviewer that we had not paid enough attention to the role of myosin IIA in the stiffness response of podosomes in our original manuscript. We have therefore performed a series of experiments to **1)** better understand the exact localization of myosin IIA within the network of actin filaments and **2)** investigate the role of myosin IIA in podosome stiffness sensing.

Revised manuscript: We now present data that myosin IIA localizes to an actin network that is approximately 500 nm higher than the ventral network and connects the top of neighbouring podosomes (**new Figure 3**). This finding does not only put forward a new concept for podosome architecture, but it also nicely puts some of our previously published results into perspective (e.g. the fact that myosin IIA inhibition does not affect vinculin localization, as reported in *Nat Comm 2013 and 2016*, is explained by the fact that vinculin and myosin IIA are enriched at different heights within the radiating actin filaments). These results are now described on **page 9-12** and discussed on **page 25** of the revised manuscript.

Subsequently, we investigated the localization and activity of myosin IIA in response to substrate stiffness. Strikingly, we did not observe any changes in both the localization as well as the activity of myosin IIA when podosomes formed on soft or stiff substrates (**new Figure 5** and **new Supplementary Figure 10**). To further support the notion that myosin IIA does not seem to play a major role in podosome stiffness sensing, we inhibited myosin IIA on both stiff and soft substrates and observed that the changes in the relocalization of vinculin are not altered (**Supplementary Figure 12**). These data are now extensively described on **page 14-15** of the revised manuscript.

Overall, these new data support a model where podosome stiffness sensing is independent from myosin IIA activity and led us to propose that podosome stiffness sensing is perhaps directly controlled by changes in actin polymerization, something we now discussed on **page 26**.

3. How the authors could measure the length of F-actin radiating filaments. In the presented figures, F-actin looks more like bundles that are densely packed in between podosomes. Could the authors describe precisely how they performed their measurements and analysis?

Response to reviewer: To measure the length of the actin filaments, we have manually tracked the ventral filaments that radiate from the core of podosomes on soft and stiff substrates. For this analysis, we have contrast enhanced the images to be sure that we tracked the entire length of the filaments.

Minor concerns:

1. WASP is not associated with the branched actin network per se, but it is rather activating the Arp2/3 complex. Thus WASP should be more associated with the membrane at the base the podosome. It could be interesting to perform SIM on Arp3 and WASP to test this.

Response to reviewer: We have not further investigated this aspect, since it has already been shown that WASP is active at the plasma membrane (as summarized by Alekhina et al. in a 2017 JCS at a glance). We therefore decided to put our effort in investigating the exact localization of myosin IIA and its role in podosome stiffness sensing since that was still largely elusive.

2. Could it be that alpha-actinin is more associated with the adhesive part of the podosome. The authors say p. 6 'α-actinin localization is markedly different from the localization of the ring component vinculin' but the diameter of alpha-actinin, at the base of the dome, does not seem different that the one of vinculin (Fig. 1e). Thus, I am not totally convinced that alpha-actinin is more associated with the core actin than with the adhesive ring. Actually, in the schematic of Fig. 2g, the authors associate pCAM with the adhesive ring. Could the authors comment on this.

Revised manuscript: We thank the reviewer for bringing this up. Indeed, the claim that we initially put forward on the differential localization of alpha-actinin and vinculin was too bold. In fact, it does make sense that alpha-actinin and vinculin partially overlap laterally. We have revisited our model and now propose that vinculin is most likely involved in anchoring the alpha-actinin crosslinked filaments to the membrane. Importantly, this blurs the concept of a classical core and ring, since the peripheral core subregion can be directly bound to vinculin. To integrate this new interpretation of the results, we have revised the cartoon in the **new Figures 2g and 3e** and rephrased the text on **page 7** of the revised manuscript. Furthermore, the novel concepts on the architecture of podosomes are now also integrated in the cartoon in the **revised Supplementary Figure 16** and discussed on **page 22 and 23** of the revised manuscript.

Reviewers' comments:

Reviewer #2 (Remarks to the Author):

Overall, this revision has substantially alleviated the reservations that I raised in the previous round of review. The data presented by the authors show that while the nanostructural organization of the core machinery of individual podosomes and the interconnected actomyosin linkages appears to be conserved across soft and stiff substrate, the ventral membrane-apposed actin compartments and vinculin appear to be mechanosensitive. However, there are a few points remaining that I feel should be addressed:

Comments:

1. The additional data on Myosin II A is valuable. However, the myosin II aspects can be considerably strengthened if a few additional characterizations are performed. In particular, the ~300 nm dimension of Myosin II, should be resolvable by AiryScan. With appropriate constructs/antibody, it should be possible to image the N- and C- terminus of Myosin IIA, by 2-color super-resolution imaging, such as shown in Myosin II studies by the works of Dylan Burnette, John Hammer, or Alexander Bershadsky. (e.g. fig.5, Burnette et al. JCB 2014). This is important since the model suggested by the author implied that the myosin II filaments should be aligned radially along the actin filaments with respect to the podosome core. With ~150 nm resolution, 2-color imaging should allow the directional organization of the 300nm filaments organization to be revealed and provide important corroboration to the model.
2. Figure 7: The title stated that Stiffness control podosome connectivity, i.e. the nearest neighbour distance. However, from the images, it would appear that the numbers of podosomes could also be affected. Other parameters should be characterized such as the total number of podosomes per cell, cell area, number of podosomes per cell area, etc., as a function of the stiffness.
 - 2.1 Figure 7b, statistical significance for cluster size comparison is missing.
3. SFIG 12: pharmacological perturbations and substrate stiffness perturbation. The type of perturbations used is relatively limited, as both Blebistatin treatment and soft substrate tend to perturb in the same direction. In line with reviewer #3's suggestion, the perturbation which increases myosin II contractility such as Calyculin A etc., should also be explored.
4. Movie 3, Fig 7j, and SFig 15A. I find it curious that the areas with the most pronounced gelatin degradation area not really under the podosomes but are instead under the focal adhesions. While a certain degree of degradation occur under podosomes, the extent of degradation is much more pronounced under the focal adhesions. This is reminiscent of Wang & McNiven, JCB 2012, which showed that focal adhesions can also be the site of gelatin degradation. Perhaps this should also be discussed, as the authors' data show that Focal-adhesion-mediated degradation appears to also be stiffness dependent.

Reviewer #3 (Remarks to the Author):

The revised manuscript by van den Dries et al., now entitled 'Modular actin nano-architecture enables podosome protrusion and mechanosensing' is improved in several ways. In this revised version, the authors explored the nanoscale distribution of myosin IIA associated with podosomes. Furthermore, the authors revealed the 3D molecular organization of actin networks connecting podosomes. These new results demonstrate that there are two super-imposed layers of actin networks, a ventral actin network associated with vinculin, and a dorsal actin network associated with myosin IIA. Changes in substrate stiffness do not impact on the architecture of the dorsal myosin IIA-associated actin network. However, the ventral actin networks and the localization of vinculin and zyxin are regulated by substrate stiffness. The length of actin structures connecting podosomes decreases with decreased stiffness, and the enrichment of vinculin and zyxin seems to increase at podosome cores, while talin localization is unchanged. Importantly, inhibition of myosin

IIA using blebbistatin on stiff or soft substrates did not change the distributions of vinculin and zyxin. Thus, all together these results strongly suggest that the ventral vinculin-associated actin network is involved in mechano-sensing, while the upper myosin II-associated actin network is not.

The authors made a significant work to address the questions raised, the answers were satisfying. In addition, the combination of previous and current results enabled the authors to build a new refined model describing the nanoscale architecture of podosomes and how this architecture is tuned by substrates stiffness. Thus I am supportive of its eventual acceptance to Nature Communications. However, I still have some remaining suggestions (described below) that could be addressed to reinforce further the manuscript, and minor comments.

1. One possibility is that local forces generated by actin polymerization on soft substrates are higher than on stiff substrates, triggering an enhanced force-dependent recruitment of vinculin and zyxin at the podosome cores. It will be important in figure 6, together with normalized fluorescence currently displayed, to measure the enrichment at podosome cores of these proteins on soft versus stiff conditions.

2. Since myosin IIA is not involved in podosome mechanosensing, it would be very interesting to test the hypothesis proposed by the authors page 26, 'A possible mechanism for podosome mechanosensing is that altered actin polymerization kinetics directly control the mechanical response of podosomes.' This could be tested using low concentrations of Latrunculin A or Cytochalasin D. This treatment should decrease the rate of actin polymerization without a complete disruption of actin networks. This might reverse the redistribution of vinculin and zyxin towards podosome cores observed on soft substrates. This will be very powerful to show in that indeed podosome mechanosensing is mainly driven by forces generated by actin polymerization and not by forces generated by myosin motors.

3. The results clearly demonstrate the differential localization of β -actin and γ -actin respectively in the central protrusion module (cPM) and peripheral protrusion module (pPM) of podosomes (Figure 2). However, the selective localization of β -actin in the lamellipodium (mainly composed of branched actin) versus γ -actin in stress fibers (mainly composed of linear actin filaments) is not obvious and not quantified (Supplementary Fig. 3a). I would remove these results from the manuscript, unless they are quantified to demonstrate selective localizations.

Minor comments:

1. Fig. 6a. It will be important to state in the text (perhaps the results section or discussion) that the actin structures in between podosomes do not correspond to individual actin filaments, but rather association of actin filaments.

2. Revealing the links between the nanoscale organization of podosomes on substrates of different rigidities and their protrusive versus degradative functions seems more difficult. This, I think may be explored in more depths in another study.

Point-by-point response to the reviewers

Reviewer #2 (Remarks to the Author):

Overall, this revision has substantially alleviated the reservations that I raised in the previous round of review. The data presented by the authors show that while the nanostructural organization of the core machinery of individual podosomes and the interconnected actomyosin linkages appears to be conserved across soft and stiff substrate, the ventral membrane-adjacent actin compartments and vinculin appear to be mechanosensitive.

However, there are a few points remaining that I feel should be addressed:

Comments:

1. The additional data on Myosin II A is valuable. However, the myosin II aspects can be considerably strengthened if a few additional characterizations are performed.

In particular, the ~300 nm dimension of Myosin II, should be resolvable by AiryScan. With appropriate constructs/antibody, it should be possible to image the N- and C- terminus of Myosin IIA, by 2-color super-resolution imaging, such as shown in Myosin II studies by the works of Dylan Burnette, John Hammer, or Alexander Bershadsky. (e.g. fig.5, Burnette et al. JCB 2014).

This is important since the model suggested by the author implied that the myosin II filaments should be aligned radially along the actin filaments with respect to the podosome core. With ~150 nm resolution, 2-color imaging should allow the directional organization of the 300nm filaments organization to be revealed and provide important corroboration to the model.

We have now performed the requested dual color experiment by labelling our samples with an antibody against the myosin light chain and an antibody against the heavy chain, which visualize the head and tail region of myosin II bipolar filaments, respectively. Imaging by Airyscan indeed reveals many myosin IIA filaments that are radially aligned with respect to the podosome core, thus oriented exactly as the radiating dorsal actin filaments, further supporting our conclusion that myosin crosslinks the dorsal actin filaments that interconnect neighboring podosomes. We have now added these new data to the main **Figure 3** and describe these new results on **page 11 and 12** of the revised manuscript. We would like to point out that to achieve the best imaging performance possible to be able to decently resolve myosin head and tail in the crowded environment of the podosomes, we have visualized myosin II light chain with Alexa488 and the myosin II heavy chain with Alexa568. This means that the actin had to be co-stained with phalloidin conjugated to a far red dye (e.g. Alexa633 or Alexa647), which is suboptimal for visualizing the very dim and thin dorsal actin filaments. Nevertheless, although the dorsal network was less well visible as compared to samples stained with Alexa488-conjugated phalloidin (Fig. 3d), the Alexa633-conjugated phalloidin overlapped well with the MHC signal, something we also show in the new **Fig. 3e** for completeness. We hope the reviewer will appreciate that the quality of our myosin Airyscan images at crowded podosomes is reasonably similar to the visualization of myosin II stacks in well-defined actin stress fibers by SIM and Airyscan in recent publications (see *Hu et al Nat Cell Biol 2017; Beach et al Nat Cell Biol 2017*). We really thank the reviewer for ‘pushing’ us to attempt this imaging as our study is now the first to provide such a detailed nanoscale visualization of myosin IIA at podosomes.

2. Figure 7: The title stated that Stiffness control podosome connectivity, i.e. the nearest neighbour distance. However, from the images, it would appear that the numbers of podosomes could also be affected. Other parameters should be characterized such as the total number of podosomes per cell, cell area, number of podosomes per cell area, etc., as a function of the stiffness.

We have now quantified the cell area and the number of podosomes per cell as a function of stiffness. This showed that for both parameters, there is no difference between stiff and soft substrates. These data are now presented in the revised **Supplementary Figure 5** and we rephrased a sentence on **page 12** of the revised manuscript to include these results. We thank the reviewer for suggesting to include these

parameters since now it is clearer that podosome connectivity is indeed specifically influenced by stiffness.

2.1 Figure 7b, statistical significance for cluster size comparison is missing.

Since the difference in podosome cluster size between cells on stiff and soft substrates was not significant as determined by a Student's t-test, we did not add statistical information. To increase clarity for the reader, we now state this in Materials and Methods (see **page 38**).

We do would like to mention two more improvements we did related to this comment. First, we noticed that the statistical information in **Figure 7d** (nearest neighbor distance) was actually lacking, something which we corrected now. Second, for consistency with the new datasets on cell area and podosome number on stiff and soft substrates, we reanalyzed the cluster size dataset. This resulted in a slightly altered graph but the conclusions remain unaltered, i.e. no difference in cluster size on stiff versus soft substrates (**Figure 7b**).

3. FIG 12: pharmacological perturbations and substrate stiffness perturbation. The type of perturbations used is relatively limited, as both Blebistatin treatment and soft substrate tend to perturb in the same direction. In line with reviewer #3's suggestion, the perturbation which increases myosin II contractility such as Calyculin A etc., should also be explored.

The reason we initially did not explore perturbations that increase myosin IIA contractility is that we know from previous studies by our group and others that this type of perturbations quickly leads to podosome dissolution (see for example *Van Helden et al JCS 2008* for stimulation with PGE2 or *Linder et al JCS 2000*, *Meddens et al Nat Comms 2016* and *Rafiq et al Nat Mat 2019* for disruption of microtubules). The issue is that myosin IIA contractility is necessary to both assemble and dissolve podosomes, indicating that myosin IIA activity must be finely regulated at podosome sites. Reproducing this situation with chemical reagents has been proven to be quite challenging in this context. Nevertheless, since we agree with the reviewer that it would be interesting to see if increased myosin IIA contractility can change the localization of vinculin at podosomes on soft substrates, we indeed tested the effect of Calyculin A on our cells and included these results in the **new Supplementary Fig. 12**. To determine the right conditions to manipulate cell contractility by Calyculin A in a controlled manner, we first performed an experiment where we tested the dose (1, 5, 10 and 20 nM) and time (5, 10, 20 and 30 min) response of dendritic cells to Calyculin A on standard glass coverslips (**Supplementary Fig. 12a**). At dosages that are reportedly effective (see *Wolfenson et al JCS 2011* and *Yam et al JCB 2007*), we observed that **1**) >10nM Calyculin A has a very detrimental effect on dendritic cells, as all cells detached within 5 minutes (data not shown); **2**) at 10 nM, cells contracted within 10 minutes and completely detached after 20 minutes; **3**) at 5 nM, cells contracted after 20 minutes dissolving most of the podosomes and detached after 30 minutes; **4**) 1 nM of Calyculin A seemed to have no effect at all. Together, the data shown in the **new Supplementary Fig. 12a**, are in line with previous observations that enhancing the activity of myosin IIA quickly results in podosome dissolution and rapid cell detachment, most likely due to the enhanced cell contraction.

Next, we tested whether vinculin localization is changed on stiff or soft substrates in response to 'gentle' myosin IIA activation by low doses of Calyculin A (1, 2.5 and 5 nM). The results from this experiment are shown in the **new Supplementary Fig. 12b** and clearly demonstrate that the localization of vinculin is not altered in response to modulation of myosin II by low doses of Calyculin A. We hope the reviewer will agree that these new data further strengthen our conclusion that myosin IIA does not appear to be involved in the relocalization of vinculin on soft substrates. These data are now described at **page 18** of the revised manuscript.

4. Movie 3, Fig 7j, and SFig 15A. I find it curious that the areas with the most pronounced gelatin degradation area not really under the podosomes but are instead under the focal adhesions. While a certain degree of degradation occur under podosomes, the extent of degradation is much more pronounced under the focal adhesions. This is reminiscent of Wang & McNiven, JCB 2012, which showed that focal adhesions can also be the site of gelatin degradation. Perhaps this should also be discussed, as the authors' data show that Focal-adhesion-mediated degradation appears to also be stiffness dependent.

As suggested by the reviewer, we added this reference in the Discussion (see **page 28**).

Reviewer #3 (Remarks to the Author):

The revised manuscript by van den Dries et al., now entitled 'Modular actin nano-architecture enables podosome protrusion and mechanosensing' is improved in several ways. In this revised version, the authors explored the nanoscale distribution of myosin IIA associated with podosomes. Furthermore, the authors revealed the 3D molecular organization of actin networks connecting podosomes. These new results demonstrate that there are two super-imposed layers of actin networks, a ventral actin network associated with vinculin, and a dorsal actin network associated with myosin IIA. Changes in substrate stiffness do not impact on the architecture of the dorsal myosin IIA-associated actin network. However, the ventral actin networks and the localization of vinculin and zyxin are regulated by substrate stiffness. The length of actin structures connecting podosomes decreases with decreased stiffness, and the enrichment of vinculin and zyxin seems to increase at podosome cores, while talin localization is unchanged. Importantly, inhibition of myosin IIA using blebbistatin on stiff or soft substrates did not change the distributions of vinculin and zyxin. Thus, all together these results strongly suggest that the ventral vinculin-associated actin network is involved in mechano-sensing, while the upper myosin II-associated actin network is not.

The authors made a significant work to address the questions raised, the answers were satisfying. In addition, the combination of previous and current results enabled the authors to build a new refined model describing the nanoscale architecture of podosomes and how this architecture is tuned by substrates stiffness. Thus I am supportive of its eventual acceptance to Nature Communications. However, I still have some remaining suggestions (described below) that could be addressed to reinforce further the manuscript, and minor comments.

1. One possibility is that local forces generated by actin polymerization on soft substrates are higher than on stiff substrates, triggering an enhanced force-dependent recruitment of vinculin and zyxin at the podosome cores. It will be important in figure 6, together with normalized fluorescence currently displayed, to measure the enrichment at podosome cores of these proteins on soft versus stiff conditions.

This is an interesting suggestion by the reviewer. To investigate a putative force-dependent recruitment of vinculin to podosomes, we have now measured the fluorescence intensity of the vinculin staining around the podosome core on stiff and soft substrates. The results clearly demonstrate that there is no difference in vinculin enrichment on stiff vs. soft substrates suggesting that local forces generated by actin polymerization may be independent from substrate stiffness. We have now added these results to **Supplementary Figure 10** and added a short description on **page 16** of the revised manuscript.

2. Since myosin IIA is not involved in podosome mechanosensing, it would be very interesting to test the hypothesis proposed by the authors page 26, 'A possible mechanism for podosome mechanosensing is that altered actin polymerization kinetics directly control the mechanical response of podosomes.' This could be tested using low concentrations of Latrunculin A or Cytochalasin D. This treatment should decrease the rate of actin polymerization without a complete disruption of actin networks. This might reverse the redistribution of vinculin and zyxin towards podosome cores observed on soft substrates. This will be very powerful to show in that indeed podosome mechanosensing is mainly driven by forces generated by actin polymerization and not by forces generated by myosin motors.

We understand the reviewer's reasoning and we have actually performed the suggested experiment several times in the past. The outcome is that a global decrease in the rate of actin polymerization by low

Figure 1 to the Reviewer. Inhibition of actin polymerization disrupts podosome organization. Cells were plated on glass coverslips for 3 hrs and stimulated with Cytochalasin D (CytoD) or Latrunculin A (LatA) at the indicated dosages and time points. Cells were subsequently fixed and stained for vinculin and actin. Shown are representative images acquired with a Leica DM widefield microscope. Scale bar = 10 μ m.

concentrations of Cytochalasin D (2.5 μ g/ml) and Latrunculin A (0.5 mM) causes an almost instant disruption of the interpodosomal network, a complete loss of vinculin and a total immobilization of zyxin (*Van den Dries et al Nat Comms 2013*). Nevertheless, in an attempt to address the reviewers' suggestion, we have set out an experiment to determine the effect of even lower concentrations of Cytochalasin D (0.25-1 μ g/ml) and Latrunculin A (0.05-0.2 mM) (**Figure 1 to the Reviewer**). The results from this experiment confirmed our previous findings that both Cytochalasin D and Latrunculin A are extremely effective in disrupting the interpodosomal actin network. Also at 1 μ g/ml of Cytochalasin D, almost all vinculin disappeared from podosomes within 2 minutes and even at dosages ten times lower than what

we previously used (0.25 $\mu\text{g}/\text{ml}$), vinculin was largely absent after 5 minutes and completely gone after 10 minutes. For Latrunculin A, we observed a similar trend although the lowest concentrations (0.05 mM) did not appear to affect podosome organization at all. From these results, we concluded that a global decrease in actin polymerization is very effective in disrupting the actin network in between podosomes and that these treatments are unfortunately far from optimal to test the proposed hypothesis. In this regard, we would like to emphasize that local actin polymerization kinetics within the cPM likely affect the organization of podosomes on stiff and soft substrates. We hope the reviewer will agree that it is extremely challenging to test this hypothesis as this would require a nanoscale manipulation/alteration of a specific set of actin filaments without altering the entire podosome architecture. Addressing this issue is a whole new project that in our view falls outside of the scope of the current manuscript. We hope that, also based on these new results, the reviewer will concur with our view. As these new experiments basically confirm previous observations, we only show them here to the reviewer to be thorough and will not include them in the manuscript. We have now changed the sentence on **page 27** of the discussion to better indicate that we propose that local polymerization kinetics could affect podosome organization.

3. The results clearly demonstrate the differential localization of β -actin and γ -actin respectively in the central protrusion module (cPM) and peripheral protrusion module (pPM) of podosomes (Figure 2). However, the selective localization of β -actin in the lamellipodium (mainly composed of branched actin) versus γ -actin in stress fibers (mainly composed of linear actin filaments) is not obvious and not quantified (Supplementary Fig. 3a). I would remove these results from the manuscript, unless they are quantified to demonstrate selective localizations.

As suggested, we have removed this Supplementary Figure and the corresponding description from the manuscript.

Minor comments:

1. Fig. 6a. It will be important to state in the text (perhaps the results section or discussion) that the actin structures in between podosomes do not correspond to individual actin filaments, but rather association of actin filaments.

We have now added the word “bundled” to the description of these filaments in the introduction on **page 3** of the revised manuscript. As such, it is clear that we do not refer to individual actin filaments when addressing the ventral and dorsal filaments.

2. Revealing the links between the nanoscale organization of podosomes on substrates of different rigidities and their protrusive versus degradative functions seems more difficult. This, I think may be explored in more depths in another study.

We fully agree with the reviewer and this is one of the research directions that we are currently undertaking.

REVIEWERS' COMMENTS:

Reviewer #2 (Remarks to the Author):

The authors have satisfactorily addressed the issues that I raised previously. I now support its acceptance to Nature Communications.

Reviewer #3 (Remarks to the Author):

The authors have added substantially to the paper and in doing so have answered most of my questions. I am happy that the authors have satisfactorily answered my concerns, and I am supportive of its acceptance to Nature Communications.